# Switchable narrow nonlocal conducting polymer plasmonics

Dongqing Lin ⬡, Yulong Duan, Pravallika Bandaru, Pengli Li,
Mohammad Shaad Ansari, Alexander Yu. Polyakov ⬡, Janna Wilhelmsen ⬡ &
Magnus P. Jonsson ⬡ ✉

Dynamically switchable surface plasmons in conducting polymers constitute an emerging route towards intelligent metasurfaces, but polymer plasmons have so far suffered from weak resonances with low quality factors ($Q < 1$-$2$). Here, we address this by nonlocal coupling of individual poly(3,4-ethylene-dioxythiophene) (PEDOT) nanoantennas through collective lattice resonances (CLR) in periodic arrays (with resonance wavelengths around 2.0-4.5 μm). The results show that careful tuning of CLR matching conditions enables organic plasmonic resonances with $Q$ up to 12. Angle-dependent extinction spectra connect the results to the enhancement of radiative coupling from diffractive lattice effects. Furthermore, the nonlocal coupling strength between nanoantenna units and lattice could be modulated via redox reactions, enabling the narrow CLRs to be reversibly switched with large modulation depth (between 7% and 45% extinction). By improving resonance strength and $Q$, the study circumvents previous limitations of conducting polymer plasmonics and shows feasibility for practical applications in active metasurfaces and nano-optics.

Conducting polymer plasmonics[1], where surface plasmon resonances originate from π-conjugated polymers, are emerging as a promising candidate in next-generation smart nano-optical devices with dynamic tuneability of light wavefronts[2] and spectral signals[1,3]. Conducting polymers such as poly(3,4-ethylenedioxythiophene) (PEDOT)[4] make organic plasmonics switchable by varying their intrinsic permittivities through the modulation of their redox states[3], distinguished from gold or silver-based inorganic plasmonic materials[5,6] with fixed permittivities. In brief, redox-tuning controls the polymer doping level and thereby carrier density and mobility, offering electrical or chemical tuning between optically metallic and dielectric response[1]. Since 2020[1,7], investigations of polymer plasmonics cover both fundamental principles and device applications, including exploration of new polymer materials[8,9], modulation of plasma frequencies and resonance wavelengths[10–12], and applications in electrically switchable metasurfaces[2,12,13]. However, previous reports on conducting polymer plasmonics were limited to localized surface plasmonic resonances

(LSPRs), which suffer from weak intensities and broad linewidths (full width at half maximum [FWHM] > 1–3 μm). This puts restrictions and performance limits for many practical applications, including meta-lenses and nanosensors[14–16]. Likewise, the low quality factors ($Q < 2$) of previous polymer plasmonics reflect low coherency and high optical losses, which is disadvantageous for nonlinear optics such as second/third-harmonic generation[17–19] or lasing actions[20–22]. By optimizing synthetic procedures[23,24] or nanocrystallization processes[25], conducting polymers may exhibit higher carrier mobilities, which should favor longer scattering times and provide narrower plasmon resonance linewidths[10,26]. Nevertheless, such improvement could be limited because of constraints related to hopping-like carrier transport in conjugated polymers[4,27].

Another strategy for high-$Q$ plasmon resonances is to transform LSPRs into nonlocal collective lattice resonances (CLRs)[6] by coupling between single nanoantennas and diffractive modes of periodic arrays[28–30]. Collecting optical energy by adjacent nanoantennas

Laboratory of Organic Electronics, Department of Science and Technology (ITN), Linköping University, Norrköping, Sweden. ✉e-mail: magnus.jonsson@liu.se

strengthens light-matter interactions[31], enables higher local density of optical states[20], enhances optical fields[32], and elongates plasmon lifetimes[33]. These features suppress radiation and dissipation losses[32], and can increase $Q$ from 5-20 (LSPR) to 100–5000 in gold nanoantenna arrays[6,15,32]. Although PEDOT nanoantenna arrays have been demonstrated to observe broad and weak LSPRs, they did not exhibit clear narrow CLR features[10,12]. An essential aspect here is that the large permittivity difference between PEDOT and a conventional metal like gold significantly affects how to achieve CLR matching conditions between the polarizabilities of single nanoantennas ($\alpha_{iso}$) and array factors ($S$)[6,34,35]. This leads to mismatch if simply exchanging gold for PEDOT under the same nanoantenna dimensions and lattice spacings. The critical development of conducting polymer CLRs for high-performance all-organic metasurfaces thereby remains. An important aspect in this respect is the potential to also modulate the coupling strength via

the polymer's redox state, aiming at offering CLR actions with reversible switching ability between "on" and "off" states.

Herein, we utilized PEDOT nanodisk antenna arrays to evolve low-quality LSPRs into more intense CLRs (Fig. 1a) with up to ten times narrower linewidths. The PEDOT used in this work was acid-treated PEDOT:Tosylate (PEDOT:Tos) made through vapor phase polymerization as detailed in the Methods section and in previous works[36]. Periodic nanoantenna arrays made from this material were fabricated through electron-beam lithography (Fig. 1b) for accurate control of nanoantenna dimensions and periodicity ($r$) (Fig. 1c). This allowed careful tuning of coupling interactions between individual nanoantennas and lattices to maximize diffractive effects and minimize damping relaxation rates, with details revealed through the dependence of periodicity and incidence angle. Optimizing the geometric matching conditions between array periodicity and nanoantenna dimensions enabled narrow conducting polymer CLRs

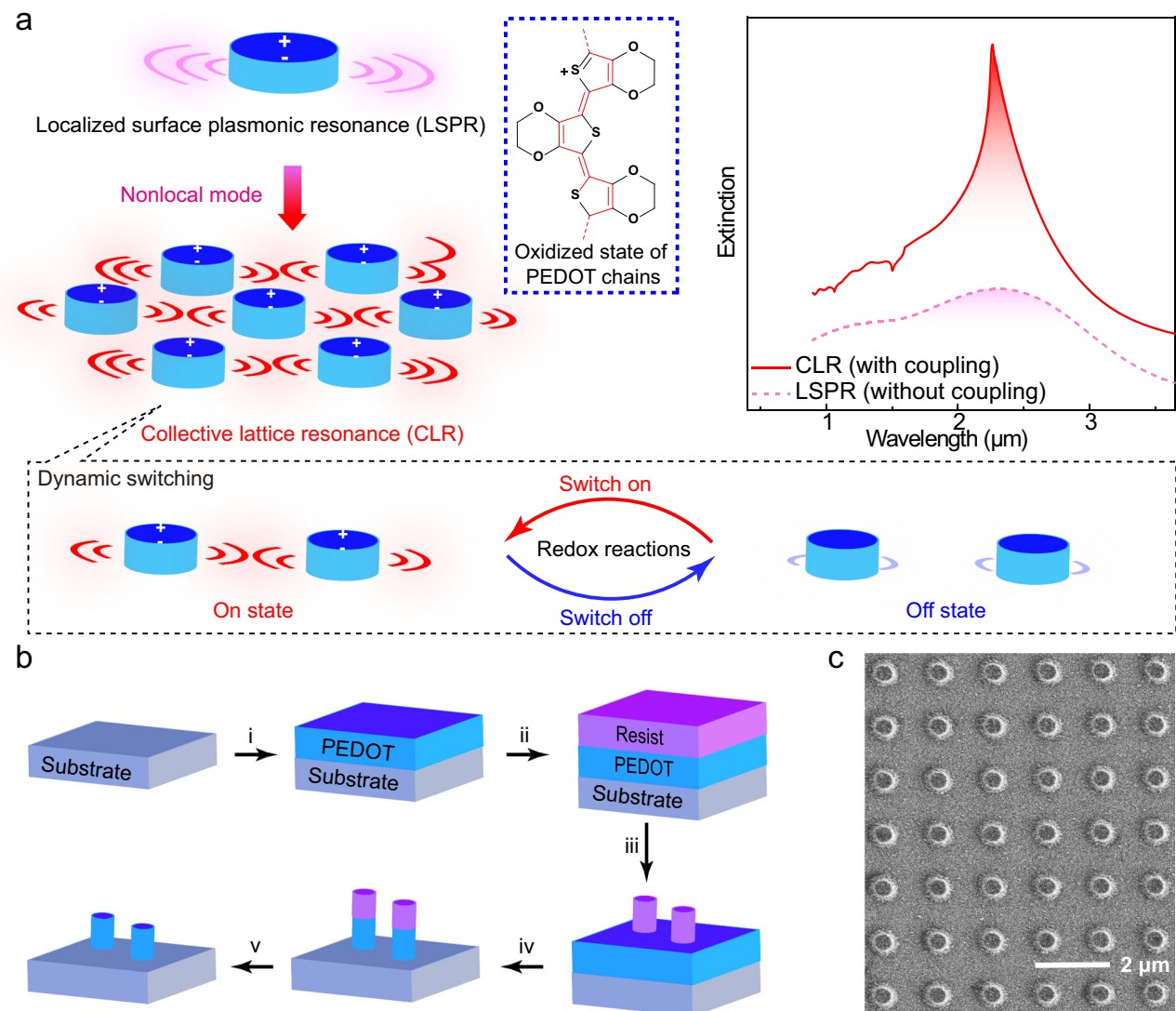

**Fig. 1 | Concept and structure of switchable collective lattice resonances with PEDOT nanoantenna arrays. a** The transformation from PEDOT nanoantenna LSPR to CLR, with dynamic switchability. The pink-color and red-color regions mark the weak (isolated nanoantenna) and intense collective (periodic array) optical response, respectively. Simulated example extinction spectra of an isolated PEDOT nanoantenna showing a broad LSPR peak (obtained without periodic boundaries) and an optimized PEDOT nanoantenna array showing CLR (obtained using periodic boundaries), are also provided in corresponding colors. The blue dashed line frame presents the molecular structure of PEDOT in its oxidized high conducting state. The black dashed frame further below illustrates the switch between "on" and "off" states via redox reactions. **b** The preparation procedure of PEDOT-based periodic arrays consisting of nanodisk units. The procedure includes: (i) preparing a layer of PEDOT film; (ii) spin-coating a layer of positive resist; (iii) electron-beam lithography, including development; (iv) dry etching with oxygen plasma; (v) removing the positive resist on top of the nanodisks. **c** SEM image of a typical final periodic PEDOT nanoantenna array.

(FWHM < 0.4–0.5 µm) across the mid-infrared wavelength range from 2.1 to 4.5 µm. We further study effects of modulating the redox-state of the PEDOT, and demonstrate switchability with large modulation depth enabled by reversible tuning between matching and mismatching CLR conditions. With $Q$ values over 10, the CLRs show a dramatic improvement compared to LSPRs from previous conducting polymer nanoantennas. The work thereby overcomes one of the main bottlenecks of conducting polymer plasmonics, showing promise for their integration toward practical applications in smart optical metasurfaces.

## Results
### Determining CLR matching conditions for PEDOT nanoantenna arrays

In periodic arrays, the polarizability of a single nanoantenna is transformed from the isolated $\alpha_{\mathrm{iso}}$ to a periodic mode $\alpha_{\mathrm{p}}$ through lattice-induced coupled dipole interactions ($S$), according to[6,34,37]:

$$\alpha_{\mathrm{p}} = (1/\alpha_{\mathrm{iso}} - S)^{-1} \quad (1)$$

$S$ quantifies the sum of coupled dipole interactions (including electrostatic and radiative couplings[38,39]) generated from other nanoantenna units in the lattice and can be evaluated using[34,37]:

$$S = \sum_{j}^{N} \exp(i\mathbf{k}r_j)\left[\frac{(1 - i\mathbf{k}r_j)(3\cos^2\theta_j - 1)}{r_j^3} + \frac{\mathbf{k}^2\sin^2\theta_j}{r_j}\right] \quad (2)$$

where $r_j$ is the center-to-center distance between two nanoantennas, $\theta_j$ is the angle between the electrical field direction and the lattice vector direction, and $N$ is the number of other nanoantennas along the specific direction. To achieve plasmon resonance with an intensified extinction $\sigma$ of the system (from $\sigma \propto |\mathbf{k}|\,\mathrm{Im}(\alpha_{\mathrm{p}})$, where $\mathbf{k}$ is the wave vector with $|\mathbf{k}| = 2\pi n_s/\lambda$, with wavelength $\lambda$ and refractive index of medium $n_s$), the CLR matching condition relates to minimizing the denominator ($1/\alpha_{\mathrm{iso}} - S$) to maximize $\alpha_{\mathrm{p}}$. At the resonance wavelength ($\lambda_r$), the CLR therefore satisfies $\mathrm{Re}(1/\alpha_{\mathrm{iso}}) = S_r$, where $\mathrm{Re}(1/\alpha_{\mathrm{iso}})$ and $S_r$ are the real parts of $1/\alpha_{\mathrm{iso}}$ and $S$, respectively[6,37]. Meanwhile, the difference between the imaginary parts of $1/\alpha_{\mathrm{iso}}$ and $S$ (denoted as $\mathrm{Im}(1/\alpha_{\mathrm{iso}})$ and $S_i$, respectively) should be sufficiently small to ensure low damping relaxation rates and to enable spectrally sharp $|\alpha_{\mathrm{p}}|$ for narrow resonances[6]. Considering corrections for the modified long-wavelength approximation[34,37], $\alpha_{\mathrm{iso}}$ of isolated PEDOT nanodisk antennas is evaluated through[34]:

$$\alpha_{\mathrm{iso}} = \alpha_s\left[1 - \frac{\mathbf{k}^2}{a}\alpha_s - \frac{2}{3}i\mathbf{k}^3\alpha_s\right]^{-1} \quad (3)$$

where $\alpha_s$ is the static non-corrected polarizability obtained by approximating the nanodisks as oblate spheroids via:

$$\alpha_s = \frac{a^2 b}{3}\frac{\varepsilon_m - \varepsilon_s}{\varepsilon_s + L(\varepsilon_m - \varepsilon_s)} \quad (4)$$

where $a$ and $b$ denote the long and short semiaxes of the oblate spheroid, respectively. $L$ is a geometric factor, and $\varepsilon_m$ and $\varepsilon_s$ are the relative permittivity of PEDOT (Supplementary Fig. 3) and the surrounding medium, respectively. Considering the inhomogeneous environment consisting of dielectric substrate ($n_s = 1.5$) and air, we applied $\varepsilon_s = 1.3$ in the calculations of $\alpha_{\mathrm{iso}}$.

Figure 2a presents the real and imaginary components of $S$ (full lines) together with $1/\alpha_{\mathrm{iso}}$ (dashed lines) for an isolated PEDOT nanoantenna (diameter [$d$] = 0.52 µm and height = 0.2 µm). To ensure efficient dipolar coupling interactions from sufficient

nanoantenna units, we set $N = 1000$ in the calculation of $S$ (Supplementary Fig. 4). The conditions that satisfy both $\mathrm{Re}(1/\alpha_{\mathrm{iso}}) = S_r$ and minor $|\mathrm{Im}(1/\alpha_{\mathrm{iso}}) - S_i|$ values are marked by purple dots, which represent the resonance wavelengths $\lambda_r$ for respective periodic distance ($r$). By increasing $r$ from 1.0 to 1.6 µm, $S_r$ increases from -6 to 33 µm$^{-3}$ at respective $\lambda_r$ (Figs. 2a, b), leading to red-shifted resonances due to dipolar coupling interactions[35,39]. Importantly, increasing $r$ moves $\lambda_r$ closer to the spectral position of the peak wavelength of $S_r$ for the respective period ($\lambda \approx n_s r$), indicating the enhancement of coupling to in-plane diffraction orders from the lattice effect[37]. We further note that the value of $|\mathrm{Im}(1/\alpha_{\mathrm{iso}}) - S_i|$ at respective $\lambda_r$ also gradually reduces with increasing periodicity, from 41 ($r = 1.0$ µm) to 17 µm$^{-3}$ ($r = 1.6$ µm). Corresponding decreased damping relaxation rate should aid resonance strength and narrowing of linewidths for larger $r$[34,35]. Figure 2c confirms this behavior, which presents $|\alpha_{\mathrm{p}}|$ for the different PEDOT arrays along with $|\alpha_{\mathrm{iso}}|$ for the isolated plasmonic PEDOT nanoantenna (also see Supplementary Fig. 6). While most arrays show $|\alpha_{\mathrm{p}}|$ peaks that are sharper and more intense compared with the isolated nanoantenna, the peak gradually increases in intensity and narrows for $r$ up to 1.6 µm. For yet larger $r$ ($r \geq 1.8$ µm), the intensity of $|\alpha_{\mathrm{p}}|$ sharply reduces at $\lambda_r \approx n_s r$, because the peak intensity of $S_r$ becomes too low to satisfy $\mathrm{Re}(1/\alpha_{\mathrm{iso}}) = S_r$ (Supplementary Fig. 7 and 8). Combined, the above results suggest that the optical extinction of CLRs based on these PEDOT nanoantennas can be sharper and more intense under optimized periodic condition $r = 1.4$–1.6 µm. The above strategy to optimize CLRs also work for other nanoantenna dimensions (Supplementary Fig. 9-16) and for other types of PEDOT[1] (Supplementary Figs. 17–19), demonstrating good generality.

Based on the prerequisite of $\mathrm{Re}(1/\alpha_{\mathrm{iso}}) = S_r$, we deduced the relationship between $S$ and normalized detuning wavelength ($\Delta$) at $\lambda_r$ for square arrays with varying periodicity (mathematic processing shown in Supplementary Note 2). $\Delta$ is used to evaluate the wavelength difference between the surface plasmon resonance ($\lambda_r$) and lattice-induced diffraction orders ($\lambda = n_s r$) via the equation[32] $\Delta = (\lambda_r - n_s r)/n_s r$. The smaller wavelength difference ($\Delta \to 0$) reflects the more significant contribution of nonlocal diffraction orders to the surface plasmon resonance, which serves as prerequisite for CLRs. $S(r, \Delta)$ mappings are shown in Fig. 2d (real part) and 2e (imaginary part). Under various periodic distances $r = 0.9 \sim 1.7$ µm, $S_r$ increases for reducing $\Delta$ from $\Delta > 0.1$ to $\Delta < 0.001$ (Fig. 2d), according to the scaling law $S_r \sim (1 + \Delta)^{-2}$ (extracted from Supplementary Equation [10]). These results confirm that the intense dipolar interactions are closely related to efficient coupling to nonlocal diffraction orders from the lattice[32]. In Fig. 2e, diminishing $\Delta$ results in more negative $S_i$ according to the scaling law $S_i \sim -(1 + \Delta)^{-1}$ (extracted from Supplementary Equation [11]), and achieves lower $|\mathrm{Im}(1/\alpha_{\mathrm{iso}}) - S_i|$, which is correlated to increasing damping relaxation lifetimes and reducing resonance linewidths. By fixing nanoantenna dimensions (with unchangeable $1/\alpha_{\mathrm{iso}}$), $\Delta$ is lower for increasing periodicity (Fig. 2d), which fulfills the scaling law $r \sim (1 + \Delta)^{-2/3}$ (extracted from Supplementary Equation [10]). Therefore, suitably large periodic distances ($r = 1.4$–1.6 µm) should be designed to access low-$\Delta$ narrow resonances.

We also evaluate hexagonal arrays of PEDOT nanoantennas to theoretically test the generality of CLR actions (Supplementary Note 2). Compared with square arrays under the same $r$, hexagonal arrays increase $S$ by ~1.5 times, owing to enhanced dipolar coupling interactions from more adjacent nanoantennas. This feature allows hexagonal arrays to fulfill $\mathrm{Re}(1/\alpha_{\mathrm{iso}}) = S_r$ under more extensive periodic distances (optimized $r$ up to 1.8 µm, Fig. 2b and Supplementary Fig. 20-23). Meanwhile, larger-magnitude negative $S_i$ of hexagonal arrays is more effective to cancel out the optical loss from damping relaxation, confirmed by lower values of $|\mathrm{Im}(1/\alpha_{\mathrm{iso}}) - S_i| = 12\text{-}14$ µm$^{-3}$ than those in square arrays ($|\mathrm{Im}(1/\alpha_{\mathrm{iso}}) - S_i| \geq 17$ µm$^{-3}$) under the individually optimized periodic conditions (Fig. 2b). In addition, hexagonal arrays with low $\Delta$ can be reached at $r = 1.6$-1.8 µm (Supplementary Fig. 20 and 22).

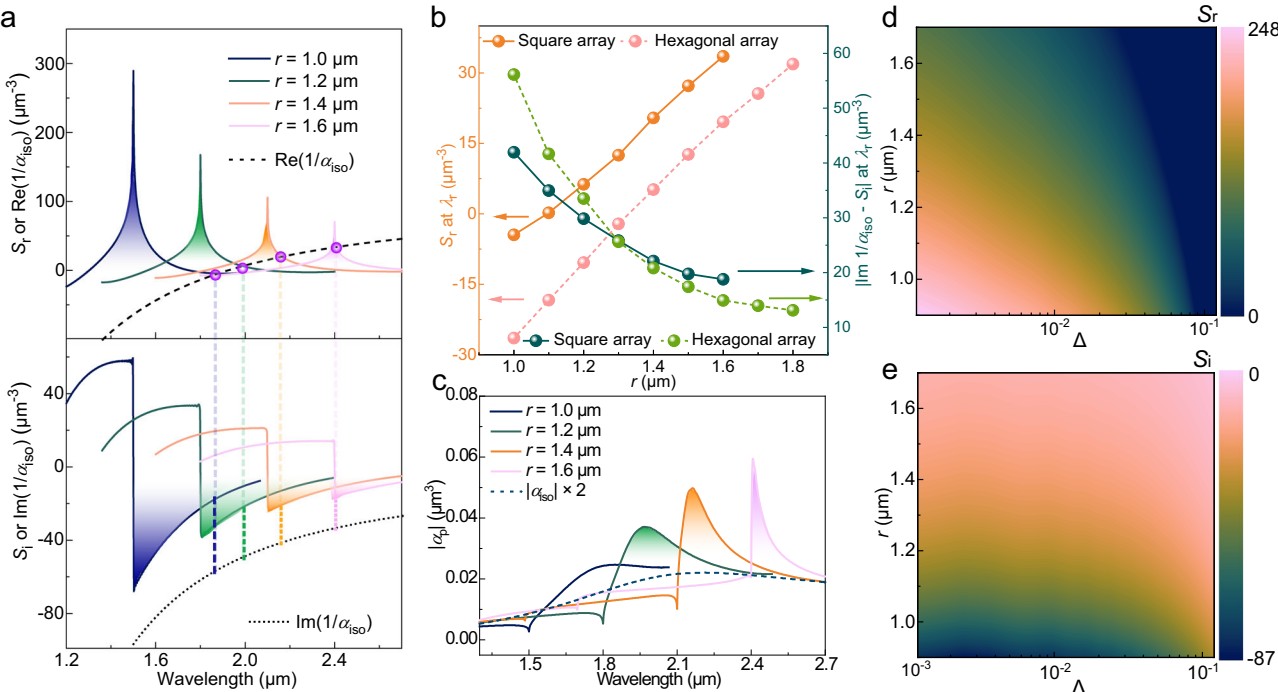

**Fig. 2 | Calculations of array factors for PEDOT nanoantenna arrays with different periodic distances. a** Analysis of the real and imaginary part of array factors ($S_r$ and $S_i$, respectively), together with $1/\alpha_{iso}$, under the periodic distance $r = 1.0, 1.2, 1.4,$ and, $1.6$ μm. $1/\alpha_{iso}$ also has a complex form, where $Re(1/\alpha_{iso})$ and $Im(1/\alpha_{iso})$ are the real and imaginary part of the reverse isolated polarizability, respectively. The length of the region with 0% transparency for each dashed line indicates the magnitude of $|Im(1/\alpha_{iso}) - S_i|$. **b** $S_r$ and $|Im(1/\alpha_{iso}) - S_i|$ at $\lambda_r$, for both square and hexagonal arrays. **c** Magnitude of periodic polarizability ($|\alpha_p|$) for PEDOT nanoantenna square arrays, along with $|\alpha_{iso}|$ from LSPR of an isolated nanoantenna. PEDOT nanodisks are set with $d = 0.52$ μm and a height of 0.2 μm. **d**, **e** $S_r$ and $S_i$ as a function of normalized detuning wavelength ($\Delta$) and $r$, at $\lambda_r$.

These results confirm the possibility of high-$Q$ CLRs also in PEDOT-based hexagonal arrays.

**Experimental demonstration of spectrally narrow conducting polymer CLRs**

To put the above analysis to experimental scrutiny, we prepared PEDOT nanodisk arrays (see scanning electron microscopy [SEM] images in Supplementary Fig. 24-25) on glass substrates ($n_s \approx 1.5$) by electron beam lithography and compare experimental extinction spectra with results from finite-difference time-domain (FDTD) simulations. Beginning with square arrays, we note that all systems exhibited clear extinction peaks that redshifted ($\lambda_r = 1.80$ to $2.32$ μm) when increasing $r$ from 1.0 to 1.5 μm (Fig. 3a). The simulations show similar results ($\lambda_r = 1.80$–$2.41$ μm, for $r = 1.0$ - 1.6 μm), consistent with more significant $S_r$ originating from dipolar coupling[35]. Importantly, the redshift is accompanied with significant narrowing of the peak linewidth from 1.0 μm ($r = 1.0$ μm) to 0.39 μm ($r = 1.5$ μm), corresponding to improvement of $Q$ from 1.8 to 6.0. FDTD simulations confirm the reduction in linewidth (from 0.9 - 1.0 μm at $r = 1.0$ μm to 0.25 μm at $r = 1.5$–1.6 μm) and show $Q$-factors enhanced to 10. We note that $\lambda_r$ for these narrow resonance peaks approaches the wavelength $\lambda \approx n_s r$, in agreement with enhancement of nonlocal coupling to lattice modes. In addition, the narrowing of resonance linewidths for redshifted $\lambda_r$ can be connected to lower radiation loss[33,35] from damping relaxation, confirmed by smaller values of $|Im(1/\alpha_{iso}) - S_i|$ at larger $r$ (Fig. 2b). Further increasing $r \geq 1.8$ μm sharply diminishes the optical extinction of CLR at $\lambda_r \approx n_s r$ (Supplementary Fig. 7), consistent with lowering of $|\alpha_p|$ (Supplementary Fig. 8) around $\lambda_r$. If the periodicity is $r = 3.0$ μm, the linewidth of the surface plasmon resonance at $\lambda_r \approx 2.1$ μm becomes much broader (FWHM = 1.1 μm and $Q \approx 1.9$), which is similar to the LSPR of single nanoantennas ($Q \approx 2$) observed by FDTD simulations

(Supplementary Fig. 27). PEDOT-based hexagonal arrays confirm the generality of forming narrow CLRs with conducting polymers (SEM images in Supplementary Figs. 28, 29). The experimental results (Fig. 3b) show clear CLRs with $\lambda_r$ redshifted from 1.78 ($r = 1.2$ μm) to 2.25 μm ($r = 1.7$ μm), and the resonance linewidth is reduced to 0.36 μm, corresponding to enhancement in $Q$-factor up to 6.5. These results are corroborated by the FDTD simulations, which suggests $Q$-factors as high as 10.

The reduction in resonance linewidth can be directly correlated with decreased $\Delta$ through the scaling law FWHM ~ $\Delta^{0.5}$, which shows good agreement when jointly plotting results for both square arrays and hexagonal arrays (Fig. 3c). These results reflect that the surface plasmon resonance from PEDOT nanoantennas is transitioned from local ($\Delta \to 1$) to nonlocal modes ($\Delta \to 0$) by increasing the periodicity. Considering $Q_d$ ~ $\Delta^{-0.5}$, where $Q_d$ is the dissipation quality factor[32], the nonlocal resonance mode diminishes the dissipation loss in the PEDOT nanoantenna arrays. In addition, minor $\Delta$ correlates with more negative $S_i$ (Fig. 2e), which suppresses the damping relaxation in LSPR and reduces the radiative loss. Therefore, low $\Delta$ favorably lowers optical losses and aids the narrow conducting polymer plasmon resonances in both square and hexagonal arrays.

So far, the study has focused on CLRs based on one particular PEDOT nanodisk diameter ($d$). We will now show that the narrow CLRs can be tailored to different spectral positions across the mid-infrared wavelength region by also allowing variation of $d$. Figure 3d shows experimental results for square (top) and hexagonal (bottom) arrays with jointly optimized $d$ and $r$ (on calcium fluoride substrate). For the square arrays, increasing $r$ from 2.4, 2.7, 2.9, and 3.0 μm and optimizing nanodisk diameters for each period ($d = 0.9, 1.08, 1.07,$ and 1.1 um, respectively, Supplementary Fig. 31), polymer CLRs with FWHM = 0.4–0.5 μm were obtained at $\lambda_r$ from 3.48, 3.88, 4.15 to 4.40

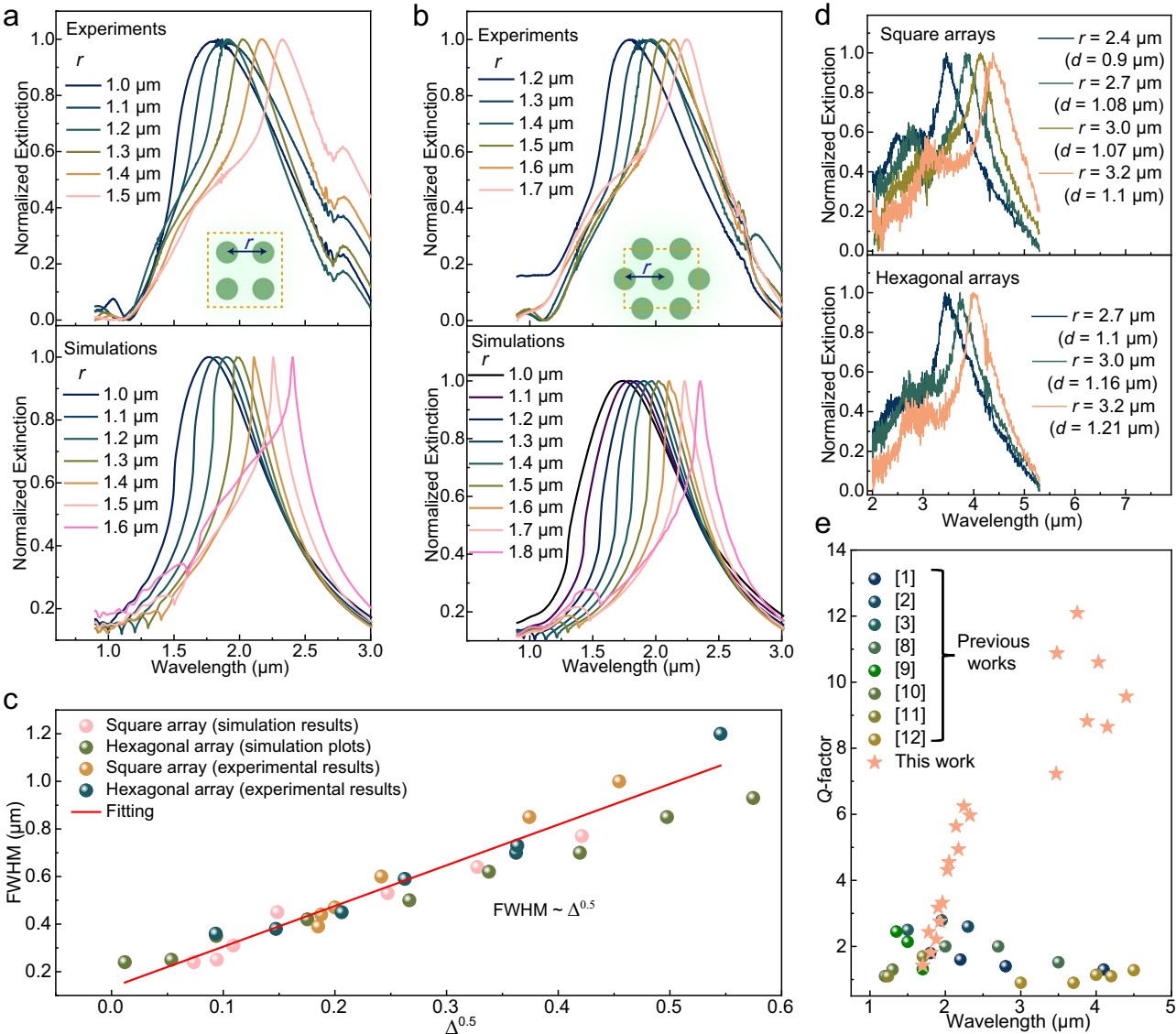

**Fig. 3 | Extinction spectra and corresponding performance parameters for PEDOT nanoantenna arrays under normal incidence. a** Experimental results and FDTD simulations of $r$-dependent extinction spectra for square arrays. The pattern of square array is shown with a periodic boundary (marked by the orange dashed line). **b** Experimental results and FDTD simulations of $r$-dependent extinction spectra for hexagonal arrays. The pattern of the hexagonal array is shown with a periodic boundary (marked by the orange dashed line). For (**a**) and (**b**), the nanoantennas' diameter (**d**) was fixed as 0.52 μm in the FDTD simulations, while the measured diameter ranged from around 0.40 to 0.50 μm for the experimental results (along with the height of ~0.2 μm). **c** Linewidths (represented by FWHM) for different experimental and simulated arrays plotted as a function of normalized detuning wavelength (Δ). **d** Extinction spectra of PEDOT nanoantenna arrays optimized for CLRs at longer peak wavelengths, with $r$ increased to 2.4–3.2 μm and $d$ increased to 0.8–1.3 μm. All PEDOT materials are based on acid-treated PEDOT:Tos. **e** Experimental $Q$-factors according to this work and previous works on conducting polymer nanoantennas[1–3,8–12].

μm (approximate to $\lambda_r \approx n_s r$ for the square arrays). The $Q$-factors for these optimized CLRs ranges from 9 to 11. Similarly, increasing $r$ for hexagonal PEDOT nanoantenna arrays (2.7, 3.0, and 3.2 μm, respectively, Supplementary Fig. 33) and optimizing $d$ (1.10, 1.16, and 1.21 um, respectively) enabled $\lambda_r$ modulation from 3.47, 3.75 to 4.03 μm. These peak wavelengths are consistent with $\lambda_r \approx 0.874 n_s r$ for hexagonal arrays. The resonance linewidths for these hexagonal CLR arrays maintained below 0.4 μm, corresponding to experimental $Q$-factors up to 12. As illustrated in Fig. 3e, the $Q$-factors reported in this study are 4-10 times higher than those in previous works of conducting polymer nanoantennas (FWHM ≥ 2-3 μm and $Q \approx 1–2$) in the mid-infrared wavelength region, the latter of which are dominated by LSPRs[1–3,8–12]. In the shorter wavelength regions ($\lambda_r < 2.5$ μm), the $Q$-factors could be enhanced by 1–4 times by forming CLRs (versus

LSPR with FWHM ≈ 0.8–1.0 μm). The concept is not restricted to the type of PEDOT that we focus on in this work, but forms a general strategy to obtain narrow resonances with optically metallic polymers (see simulation results in Supplementary Fig. 35).

**Understanding conducting polymer CLRs via angle-dependence**
To determine the mechanisms for nonlocal coupling in PEDOT nanoantenna arrays, we study the effect of varying the incident angle ($\theta_i$). The incident light was polarized in transverse-magnetic (TM) mode (p-polarization, see Fig. 4a, b). As illustrated in Fig. 4b, the coupling with (±1, 0) Rayleigh anomaly (RA) can be assigned to the diffraction grating effect that obeys the equation[31] $\lambda_r = r (n_s \pm \sin \theta_i)$ along the X-axis. Along the Y-axis, the coupling with (0, ±1) RA that follows the equation $\lambda_r^2 = r^2 (n_s^2 - \sin^2 \theta_i)$ can be assigned to the dipole

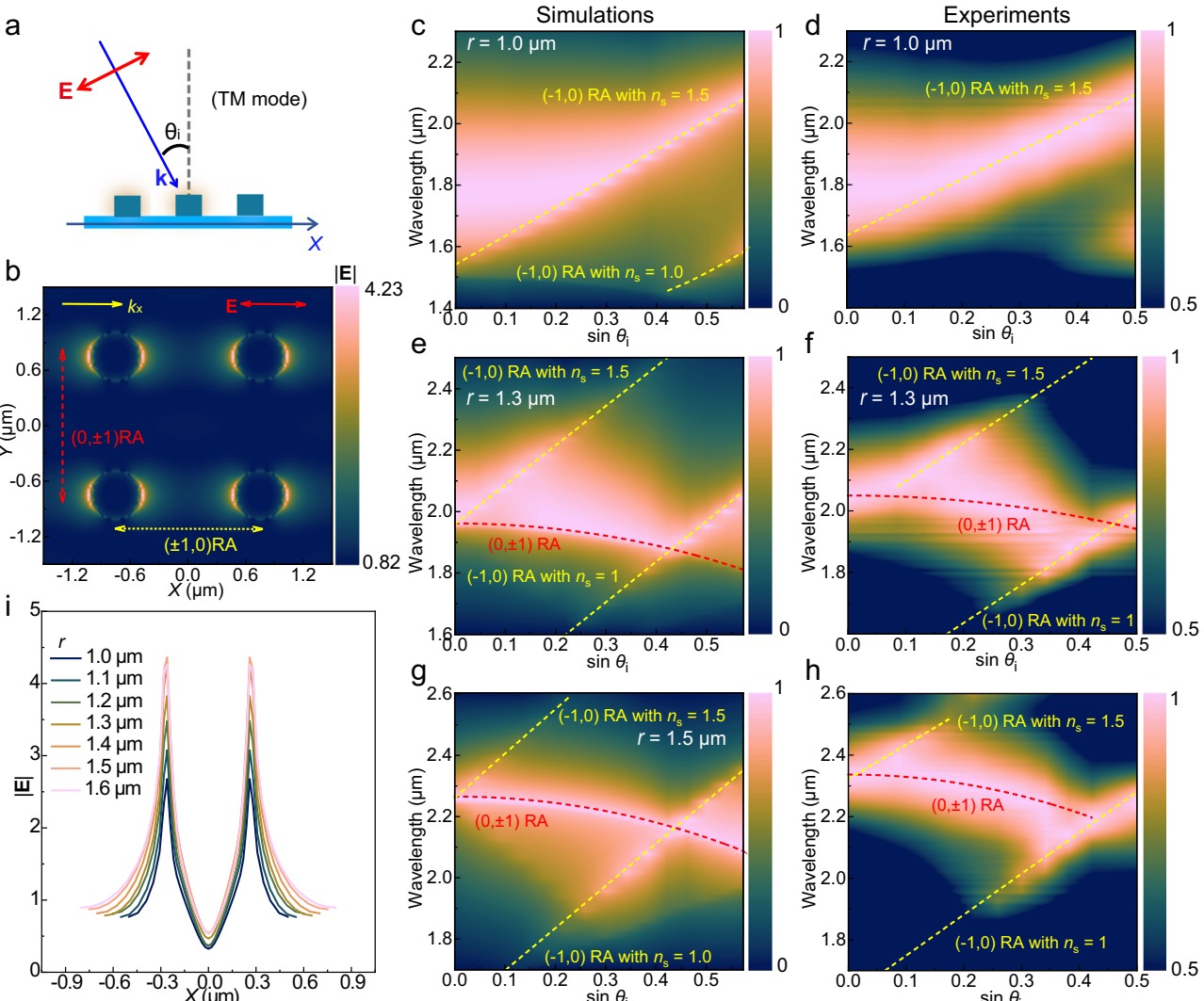

**Fig. 4 | Angle-dependent optical response of PEDOT nanoantenna arrays.**
**a** Illustration of the experimental conditions based on the transverse magnetic (TM) mode, where the orientation of the electrical field (**E**) and wave vectors (**k**) and incident angle ($\theta_i$) are also defined. **b** Illustrations of the in-plane components of the electric field polarization and wave vectors (along the X-axis, marked as $k_x$) from the incident light, and the diffraction orders of Rayleigh anomaly (RA) based on the electrical field distribution of square arrays, as calculated by FDTD simulations. **c**, **e**, and **g** Simulated angle-dependent extinction spectra (normalized) for PEDOT nanoantenna square arrays with $r = 1.0$, 1.3, and 1.5 μm, respectively. The experimental results (normalized extinction) with $r = 1.0$, 1.3, and 1.5 μm, are shown in (**d**, **f**, **h**), respectively. The red dashed line represents the RA originating from dipolar radiation, and the yellow dashed lines are the RAs derived from diffraction grating effect based on refractive index of medium ($n_s$). **i** Electric field distribution (denoted by the magnitude |**E**|) of square arrays with different $r = 1.0$–1.6 μm at $\lambda_r$, where the nanoantenna was centered at ($X = 0$, $Y = 0$). For **b** and **i**, the $Z$ coordinate for the electric field distribution was set at the interface between the PEDOT nanoantennas and the dielectric substrate. The diameter of the PEDOT nanoantennas was set to 0.52 μm for all panels (≈0.5 μm in the experiments).

radiation effect[33], as the long-range coupling interaction perpendicular to the electric polarization. Figure 4c-h show simulated (Fig. 4c, e, g) and experimental (Fig. 4d, f, h) angle-dependent extinction spectra for PEDOT nanoantenna arrays with constant $d$ (0.52 μm for simulations and ~0.5 μm also for the experiments, Supplementary Figs. 24, 25) and increasing $r$ as detailed below. In Fig. 4c, d, the smallest array period ($r = 1.0$ μm) shows the least clear CLR at normal incidence among the studied systems. Under small incident angles ($\theta_i \leq 10°$), the array structure exhibits an angle-independent broad feature. Such non-dispersive mode reflects low coupling interactions from lattice effects, similar to LSPR[31], consistent with low contribution from $S_r$ and high Δ. Further increasing $\sin\theta_i$ enables the introduction of $(-1, 0)$ RA coupling (on glass substrate $n_s = 1.5$) from diffraction grating effects, which facilitates the transformation from LSPR into a clear but weak CLR. However, $(0, \pm 1)$ RA coupling from dipole radiation is not observed

even for larger angles, in agreement with a large $|\mathrm{Im}(1/\alpha_{\mathrm{iso}}) - S_i|$ that indicates large damping relaxation. These features without efficient radiative coupling are also observed in periodic arrays with $r = 1.1$ μm (Supplementary Fig. 36).

When increasing $r$ to 1.3 μm (simulation results in Fig. 4e and experimental results in Fig. 4f), the resonance peak is split into two dispersive modes upon increasing the incidence angle, where one mode has the coupling with $(-1, 0)$ RA ($n_s = 1.5$) from diffraction grating effect and the other mode exhibits coupling with $(0, \pm 1)$ RA from dipole radiation. At even larger $\sin\theta_i = 0.4$–0.5, such radiative $(0, \pm 1)$ RA coupling interacts with the $(-1, 0)$ RA ($n_s = 1.0$, in the air), which generates an additional sharp resonance. These results verify the enhancement of nonlocal diffraction orders in plasmonic resonances when the CLR matching conditions are combined with moderate $S_r = 12$ μm$^{-3}$ and Δ = 0.04. As expected, the above coupling features are

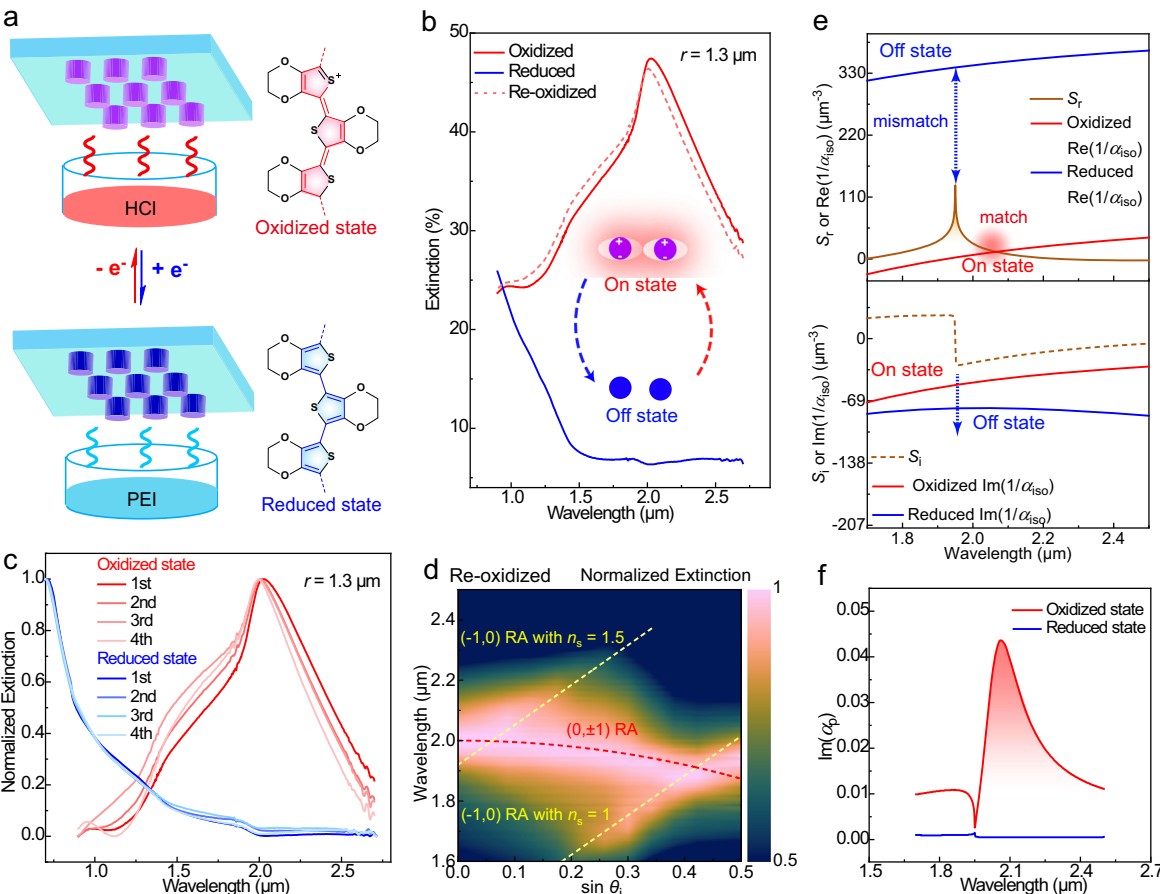

**Fig. 5 | Redox switching of PEDOT nanoantenna CLRs. a** Illustration of the redox modulation process of PEDOT nanoantennas. The reduced state is formed by PEI vapor treatment while the oxidized state could be retrieved via hydrochloric (HCl) vapor treatment. **b** Measured extinction spectra of a PEDOT-based periodic nanoantenna array in the oxidized state (on), reduced state (off), and re-oxidized state after one full redox-cycle (on). **c** Measured extinction spectra (normalized) of the PEDOT nanoantenna array in the oxidized (during the first redox cycle) and re-oxidized states (during the 2nd-4th redox cycle), along with the reduced state (during the 1st–4th redox cycle). **d** Measured angle-dependent extinction spectra (TM mode) of the PEDOT nanoantenna array in the re-oxidized state during the 4th

redox cycle. **e** Analysis of $(1/\alpha_{iso})$-$S$ relationships for both oxidized and reduced states of the PEDOT nanoantenna array, including both the real (Re[$1/\alpha_{iso}$] and $S_r$) and imaginary (Im[$1/\alpha_{iso}$] and $S_i$) components. The calculations of oxidized and reduced $1/\alpha_{iso}$ are based on the permittivity of the PEDOT material in the oxidized and reduced states, respectively. The red shaded point represents the matching conditions for CLR in the on state. **f** The imaginary part of the periodic polarizability (Im[$\alpha_p$]) of the PEDOT-based periodic array in the different redox states, as obtained by calculations through Eq. (1). For both experimental results and theoretical calculations, the PEDOT nanoantenna units are on glass and have a diameter of 0.5 μm and a height of 0.2 μm in an array structure with periodicity $r = 1.3$ μm.

also observed for the PEDOT nanoantenna array with optimized period ($r = 1.5$ μm, simulations in Fig. 4g and experimental results in Fig. 4h). The peak associated with diffraction orders from dipole radiation is more intense than that from diffraction grating effects. This enhanced radiative coupling is mainly attributed to slow-decaying radiation fields, which agrees with further decrease of |Im($1/\alpha_{iso}$) − $S_i$| when increasing $r$ from 1.3 μm to 1.5 μm. To confirm the contribution of dipole radiations to the CLR actions, we investigate the electric field distributions of the PEDOT nanoantenna arrays (at $\lambda_r$) via FDTD simulations. At the hotspot positions of the PEDOT nanoantennas (at $X = \pm 0.26$-$0.28$ μm in Fig. 4i), increasing $r$ from 1.0 to 1.6 μm gradually enhances the magnitude of the electric field |**E**|, which is also observed in hexagonal arrays (Supplementary Fig. 39). These results reflect the intensified dipole radiations due to the lower |Im($1/\alpha_{iso}$) − $S_i$| (Fig. 2b) and the enhanced |$\alpha_p$| (Fig. 2c) for the arrays with larger $r$. However, if the periodicity is $r = 3.0$ μm, the broad peak at $\lambda_r \approx 2.1$ μm does not show the type of angle-dependent features that would be required for either diffraction grating effect or lattice dipole radiation coupling (Supplementary Fig. 40). This confirms that such large periodicity for the same nanoantenna dimensions leads to optical response

dominated by LSPR, without major contribution from lattice effects. Localized resonance modes were also observed if limiting the lattice size [($N + 1$) × ($N + 1$)] to only allow coupling to the closest neighboring nanoantennas ($N = 1$, Supplementary Fig. 41). We estimate a lattice size of $N \approx 150$ to be sufficient to support efficient nonlocal resonance with intense coupling to dipole radiation (Supplementary Fig. 42). Therefore, effective radiation coupling from suitably periodic long-range lattices is key in narrow conducting polymer CLRs, similar to gold-based systems[33].

## Redox-switching of the CLR matching conditions

We further demonstrate switching of the PEDOT-based CLR actions by modulating the redox-state of the polymer. While such tuning is known to affect the response of individual nanoantennas, effects here also involve modulation of the CLR matching conditions. We selected a square array with $r = 1.3$ μm ($\lambda_r = 2.03$ μm) of PEDOT nanoantennas with $d = 0.5$ μm and a height of 0.2 μm. Figure 5a illustrates how to switch the redox state of PEDOT by oxidation with hydrochloric (HCl) acid vapor[40] or by reduction with poly(ethylenimine) (PEI) vapor[1,40], where the latter diminishes π-electron delocalization and decreases

the electrical conductivity by at least three orders of magnitude. In the oxidized high-conducting state of the PEDOT, the nanoantenna array provides a narrow CLR extinction peak at 2 µm, with peak extinction of around 45% and FWHM = 0.43 µm (Fig. 5b). When reducing the material by exposure to PEI vapor for 5 min (under nitrogen atmosphere), this intense plasmonic CLR peak is eliminated, which can be denoted as the "off state" of the CLR. Acid treatment of the reduced PEDOT nanoantenna array then enables the regeneration of the CLR to its "on state". The modulation depth of the extinction is up to 40 percentage points between the oxidized (≈45% extinction) and reduced (≈5% extinction) states, corresponding to around 2 times larger modulation depth than for periodic arrays without CLR in previous works (20–25 percentage points)[2,12]. Apart from extinction intensities, narrow resonance linewidth is also observed (FWHM ≈ 0.37–0.45 µm) after re-oxidation. Even under 3–4 redox cycles, we still observe the disappearance and re-appearance of the intense and narrow plasmonic CLR (Fig. 5c), with a small (~10 nm) blue-shift after each redox cycle as discussed more below. Angle-dependent measurements confirm that the CLR in the re-oxidized state (during the 4th cycle, Fig. 5d) still shows efficient radiative coupling with (0, ±1) RA and the diffraction grating effect with (−1, 0) RA. The generality of redox-switchable conducting polymer CLRs is shown through another array structure with a longer periodic distance $r = 1.5$ µm (Supplementary Fig. 48).

To understand the mechanism of the switchable CLR response, we explore changes in the relationship of $(1/\alpha_{iso})$−$S$ between the oxidized and reduced states of the PEDOT nanoantenna array. Since the periodic distance is not modulated during redox cycling, $S_r$ and $S_i$ should be the same in the oxidized ("on state") and reduced state ("off state"), according to Eq. (2). By contrast, the transformation from the oxidized to the reduced state enables drastic variations in $1/\alpha_{iso}$ through permittivity modulation, because reducing the material to a low-conducting state drastically decreases the imaginary part of the permittivity and also switches sign of the real permittivity from negative to positive (Supplementary Fig. 49). According to Eqs. (3) and (4), the significant rise of $Re(1/\alpha_{iso})$ for the reduced PEDOT nanoantennas generates a large mismatch gap by $|Re(1/\alpha_{iso}) - S_r| \approx$ 210 µm⁻³ at the $S_r$ peak ($\lambda = n_s r = 1.95$ µm, in Fig. 5e). This result suggests that the dipolar coupling interaction from $S_r$ becomes insufficient to eliminate $Re(1/\alpha_{iso})$, making it impossible to fulfill the CLR matching condition (Fig. 5e). Furthermore, the more negative $Im(1/\alpha_{iso})$ of the reduced PEDOT nanoantennas enlarges the value of $|Im(1/\alpha_{iso}) - S_i|$ by ~3 times at $\lambda = 1.95$–2.03 µm, indicating more serious damping relaxation. Figure 5f shows that these two changes in $1/\alpha_{iso}$ leads to vanishing of the sharp peak of $Im(\alpha_p)$ in the reduced state (in the "off state"). Meanwhile, because the real permittivity is not negative around 1-3 µm, we also do not observe the LSPR signal of the individual nanoantennas in the reduced state. The above analysis is consistent with simulated extinction spectra (Supplementary Fig. 50) based on the permittivity of the reduced state.

When returning to the high-conducting oxidized state via acid treatment, $Re(1/\alpha_{iso})$ is reduced approximately to the original "on state" so that dipolar coupling interactions can again cancel out $Re(1/\alpha_{iso})$ at $\lambda_r$, which favorably re-generates the intense $Im(\alpha_p)$ peak and re-excites the CLR ("on state"). Moreover, $Im(1/\alpha_{iso})$ becomes less negative to diminish differences from $S_i$, which allows the recovery of low-loss radiative coupling to ensure the narrow linewidth and high intensity of the re-generated CLR. The small changes observed for the "on state" between cycles may be associated with slightly lower conductivity of the re-oxidized state, leading to marginally larger $Re(1/\alpha_{iso})$ which slight blue-shifts the resonance peak and lowers $\Delta$, while the CLR matching conditions can be still fulfilled for the "on state". We conclude that the redox-induced switching of the $(1/\alpha_{iso})$-$S$ relationship allows for large modulation of intense conducting polymer CLRs.

## Discussion

Our study improves the $Q$-factor of conducting polymer plasmonics from typical values below 2 to values up to above 10, by nonlocal coupling of individual nanoantennas to CLRs. Optimizing geometric parameters including array periodicity and nanoantenna dimensions enable extinction peak linewidths <0.4 µm in the mid-infrared wavelength region of 2–4.5 µm, with experimental $Q$-factors up to 12. Angle-dependent extinction spectra show that efficient radiative coupling from diffractive lattice structures dominates the narrow CLR action. Furthermore, redox-modulation can tune the PEDOT nanoantenna arrays between matching and mismatching the CLR conditions, leading to on/off switching with extinction modulation depth as large as 40 percentage points. The work takes important steps for conducting polymer plasmonics by overcoming the major challenges of weak and broad resonances and providing performance values suitable for practical applications.

## Methods

### Thin-film deposition

The PEDOT nanoantennas were prepared based on the acid-treated PEDOT:Tosylate (PEDOT:Tos) thin films using the procedure described in previously reported literature[36,41]. Briefly, precleaned glass substrates were spin-coated with the oxidant reagent (the mixture of 2 g of tri-block PEGPPG-PEG co-polymer (from Sigma Aldrich company), 2 g of Clevios C-B 54 V3 (from Heraeus company in German), and 5 g of absolute ethanol) with a spinning speed of 1500 RPM, followed by annealing at 70 °C for 1 min. The oxidant-coated films were then placed in a vacuum chamber for vapor phase polymerization (VPP). In this step, the oxidant-coated films were exposed to EDOT vapor, generated by heating liquid EDOT at 60 °C under vacuum, and then reacted with EDOT vapor for 40 min. Then, films were rinsed with ethanol to remove any unreacted residues and subsequently dried using a gentle flow of nitrogen gas. The resulting VPP films with a thickness of about 0.2 µm were produced. For acid treatment, we soaked these PEDOT films in 3 M sulfuric acid solution for 10 min at room temperature and then washed these samples with deionized (DI) water, followed by nitrogen flow drying and annealed at 140 °C for 10 min, which can improve the electrical conductivity of PEDOT films. If using CaF₂ substrates, the VPP conditions and procedures were the same as those using glass substrates. Still, during the acid treatment, we use 1.5 M sulfuric acid solution containing 50% volume ratio of ethanol, because the 3 M sulfuric acid solution can easily separate PEDOT film from CaF₂ substrates.

### Periodic array fabrication and characterization

The acid-treated PEDOT:Tos films on glass substrate were covered with ZEP520A positive resist (ZEONREX Electronic Chemicals) using spin-coating at 2000 rpm. We baked the resist at 130 °C for 3 min. The resist thickness was 0.5–0.6 µm. Then the specific regions for periodic arrays were exposed using Raith Voyager 100 electron beam lithographer operating at 50 kV. We used beam current of 1.6–1.7 nA and set an area dose and curved elements dose to 113 µC cm⁻². We also optimized pattern design to reach the target size of the nanodisks. We did not deposit any additional conductive layer because the PEDOT:Tos films provide good charge dissipation during e-beam lithography. After competing exposure, we developed the resist for 75–90 s using ZED-N50 developer (ZEONREX Electronic Chemicals) and blow-dried the samples. Then we dry-etched PEDOT:Tos film through ZEP520A mask by reactive oxygen plasma in a RIE Vacutec etcher. We etched at 150 mTorr pressure and 50 W forward power for 155 s to reach slight over-etching of the areas around the nanodisks. Finally, we soaked the etched sample into the remover AR 300-72 at 60 °C for 3 min to remove the remaining positive resist from the top of PEDOT:Tos nanoantennas. The final periodic structures were visualized without

any additional coating using Carl Zeiss Sigma 500 scanning electron microscope operating at 1 kV voltage with SE2 detector.

## Redox-recycles

The chemical redox reactions were referred from previous works[1,40]. In brief, the reduction reaction was performed via the vapor of branched poly(ethylenimine) ($M_w \approx 800$, from Sigma–Aldrich company) in a nitrogen-filled glove box (to ensure the full effectiveness of the reduction reaction with PEDOT). The reduction reaction was carried out under 120 °C for 5 min, followed by annealing at 120 °C for 10 min. For the re-oxidized state of PEDOT-based periodic arrays, we performed the oxidation reactions with the HCl vapor (Sigma–Aldrich, 37%) under 20 °C for 15–20 min. It is noted that although 3 M sulfuric acid solution can also induce oxidation reactions via soaking the sample, but such solution can easily destroy the periodic arrays.

## Extinction spectra characterization

The extinction spectra were recorded using UV–Vis–NIR spectrophotometer (Perkin Elmer Lambda 900) in the wavelength regions of 0.9–3 μm (when using glass substrates. The angle-dependent spectra were collected through the angle-resolution transmission accessory, and the aperture (a diameter of 5 mm) and polarizer are also used. As the spectra intensity becomes much lower when additionally using an aperture and a polarizer, we performed the smoothing process for extinction spectra to testify the resonance peak. The extinction spectra in the longer-wavelength region of 2–6 μm were based on the transmission characterization via FTIR machine (PerkinElmer Spectron 3), when using $CaF_2$ substrates. During all above extinction characterizations, the sample of PEDOT nanoantenna array was placed in air.

## Electrical conductivity characterization

The sheet resistance $R_s$ of PEDOT films was measured through a four-point probe machine consisting of a Keithley 2400 part and a Signatone Pro4 S-302 resistivity stand. The surface profiler (Dektak XT Bruker) is used to evaluate the acid-treated PEDOT:Tos film thickness ($h$). The electrical conductivity was evaluated based on the equation $\sigma = 1/(R_s h)$.

## Ellipsometry characterization

Two different ellipsometers, each with different spectra ranges, were used to collect the ellipsometry data of the acid-treated PEDOT:Tos film and the PEI-reacted PEDOT film on silica substrates (with an ultrathin $SiO_2$ layer of 1 nm on the top of silica layer). All ellipsometry measurements were conducted at 20 °C. For the wavelength range of 0.4–1.69 μm, the UV–vis–NIR measurements were carried out with the J. A. Woollam Co. RC2 spectroscopic ellipsometer using four incident angles (45°, 55°, 65°, and 75°). For the wavelength range within 1.69–10 μm, infrared measurements were based on the J. A. Woollam Co. IR-VASE spectroscopic ellipsometer using three incident angles (50°, 60°, and 70°). We used the CompleteEASE (J. A. Woollam Co.) software to fit and analyze the refractive index based on the anisotropic Drude–Lorentz model[41], and the parameters in Drude-Lorentz models such as frequencies, damping parts, and amplitudes were further confirmed by Matlab software.

## Optical numerical simulations

FDTD simulations were based on the Ansys Lumerical FDTD software (https://www.ansys.com/products/optics/fdtd) to achieve far-field (extinction spectra) and near-field (electrical field distribution) optical response of PEDOT-based periodic arrays. The PEDOT-based periodic arrays were constructed with periodic boundaries (Bloch type) along the X-axis (parallel to the electric field polarization direction) and the Y-axis. By contrast, PML boundaries were used along Z-axis (as parallel to the direction of wave vector from incident light). A dielectric material with the constant refractive index $n_s = 1.5$ was set as the

substrate. The refractive index of the PEDOT nanodisk units (with a height of 0.2 μm and $d = 0.5 - 0.52$ μm) was based on the anisotropic Drude–Lorentz model analysis from the experimentally obtained ellipsometry data. We used a Bloch/periodic type plane wave to illuminate the nanoantenna at normal incidence along the Z-axis from air medium. For angle-dependent simulations, we used a BFAST type plane wave as the light source. Power monitors were used to record the far-field and near-field distribution profiles. The mesh size was given as $20 \times 20 \times 10$ nm$^3$. For the localized model consisting of a nanoantenna (a diameter of 0.52 μm) on the dielectric substrate ($n_s = 1.5$), we used PML boundaries along all directions and selected Total-Field Scattered-Field (TFSF) source as the light source. The cross-section spectra of LSPR were achieved by box-like power detectors.

## Data availability

All data generated in this study is provided in the Supplementary Information and/or available at Zenodo at https://doi.org/10.5281/zenodo.15100375.

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

## Acknowledgements

The authors acknowledge support from the European Research Council (Consolidator Grant, 101086683), the Knut and Alice Wallenberg Foundation (Wallenberg Academy Fellow, 2019. 0163), the Swedish Research Council (VR, Consolidator Grant, 2020-00287), and the Swedish Government Strategic Research Area in Materials Science on Functional Materials at Linköping University (Faculty Grant SFO-Mat-LiU No. 2009 00971).

## Author contributions

M.J. and D.L. conceived the study of conducting polymer CLRs. D.L. prepared acid-treated PEDOT:Tos films with the assistance of M.S.A., fabricated periodic arrays with the assistance of A.P., calculated CLR conditions after discussion with Y.D., acquired extinction spectra with the help of Y.D., M.S.A., and, P.L., performed FDTD simulations and redox reactions with the assistance of P. B., performed ellipsometry measurements and analyzed refractive index data with the assistance of P.L. SEM characterizations are accessed with the help of J.W. The significant idea of hexagonal arrays is provided by Y.D. M.J. supervised the project. This manuscript is written by D.L. and M.J. The project and results were regularly discussed among M.J., D.L., and, Y. D.

## Funding

## Competing interests

The authors declare no competing interests.
