## [Transparent Peer Review file · Nature Communications]

Switchable Narrow Nonlocal Conducting Polymer Plasmonics

Corresponding Author: Professor Magnus Jonsson

Version 0:

Reviewer comments:

Reviewer #1

(Remarks to the Author)

This is an interesting article in a fast-moving area within nanophotonics. It merits publication since it provides topical new results that show how appropriate nanostructure can be used to improve the plasmonic performance of conducting polymers. In general the work is very well presented. There are a number of questions that the authors should consider:

Main text

- 1) The lattice resonances that are considered here usually require an homogeneous (refractive index) environment if they are to be observed. Maybe I missed this, but I could not see in the manuscript whether an homogeneous environment had been used. If YES, then this needs to be stated, including the technique used; If NO, then do the authors have an explanation for why they see the resonances that they observe, despite the lack of a homogeneous environment?
- 2) It would be an improvement if the authors mentioned the spectral region of interest sooner, ideally in the abstract, and earlier in the introduction.
- 3) There are numbers missing from the axes in fig 1a (whilst the figure is illustrative, it would help the reader if they specified the spectral range)
- 4) When describing the enhancement in Q, it might be better to simply state the best Q obtained (12 I think) instead of "enhanced up to 10 times", "by around one order of magnitude" (conclusion).

Supp Info

- 1) Lorentz oscillator parameters are given in many places. Please also provide the relevant Lorentz oscillator equations. If this is done then others can replicate the work.
- 2) A suggestion. On p.3, it seems odd that there is only one Drude component in-plane, whilst out-of-plane there are two Lorentz oscillators, at 0.025 eV and 0.028 eV, and these two oscillators seem to do the same job. Could these two out-of-plane LOs be replaced with a (single) Drude component?
- 3) It looks as though the authors are trying to fit over the spectral region 0.14-5.8 eV, and yet have multiple oscillators with energy < 0.14 eV and multiple oscillators with energy > 5.8 eV. It's not unusual to have an oscillator beyond the region to which a fit is to be produced, however, could they achieve a similar fit with just one oscillator at each end of the spectrum?
- 4) The fit in the region (up to 5um) in supp figure S2 is not as good as the excellent fit in the NIR in supp figure S1; in particular the Psi spectra seem to be off by a couple of degrees, which in turn means that the optical constants will be slightly out. I don't know if it's enough to worry about, but won't a small change in optical constants have a bit of a knock-on effect in their coupled dipole calculations?

Reviewer #2

(Remarks to the Author)

This is a very interesting paper which clearly demonstrate that resonance strength and quality factor of conducting polymer plasmonic can be strongly improved through collective lattice resonance.

The results are, in my opinion, very important and original. They solve a major bottleneck of organic plasmonics and widens the practical applications of organic plasmonics as building blocks of active metasurfaces and nano-optic systems.

The manuscript is well written but I suggest to define more carefully the key parameters such as S (page 3) and δ

(page 5) as many readers will not be specialized enough in lattice effect and detuning wavelength to easily follow the simulation page 4 and 5.

Experimental results are impressive and will be of interest to many scientists reading nature communications.

I strongly recommend publication in nature communication

Reviewer #3

(Remarks to the Author)

The manuscript titled "Switchable Narrow Nonlocal Conducting Polymer Plasmonics" describes how using a regular array (in either a hexagonal or square lattice) can increase the otherwise poor Q-factors of conducting polymer plasmonics. They also demonstrate how switching PEDOT between redox states can be used to switch on and off the plasmonic properties of the arrays.

The paper is well structured, and nicely combines extensive simulations and experiments. They show how the diameter and lattice spacing can be tuned to optimise the Q-factor of the structures, allowing q-factors to be reached around 10-12 at $\sim 4\mu\text{m}$.

One of the notable results is the increase in q-factor as a result of the array though both the number of nearest neighbours and the ordered lattice is discussed. Since the hexagonal lattice and square lattice provide roughly similar results and the hexagonal lattice does a little better since it has "more adjacent neighbours" It would be interesting to disentangle local effects from periodic lattice effects. Can the author think of a way of disentangling adjacent neighbour effects from periodic lattice effects, e.g. what would happen in a similarly spaced but disordered array? How large does the lattice need to be to get this benefit?

They demonstrate the switching of the PEDOT arrays using either PEI or HCL vapor. In figure 5c they show the switching result for multiple REDOX cycles. Can the authors also include the off (reduced) state for each cycle in this figure?

In the synthesis part it is not clear what is meant by: "the oxidant-coated films were exposed to EDOT vapors, generated by heating liquid EDOT at 60 °C, and then reacted with EDOT for 40 min" Is that a EDOT solution, are any steps involved in between? or just dipped in pure EDOT? Is this based on a literature method?

Reviewer #4

(Remarks to the Author)

Version 1:

Reviewer comments:

Reviewer #1

(Remarks to the Author)

The authors have addressed the questions we raised and we are satisfied with their response.

Reviewer #2

(Remarks to the Author)

This is a very interesting paper which brings new and original developments in Organic plasmonics

The results are important and open new perspective for these conductive polymer nanostructures.

All the reviewers comments have been well answered in the revised manuscript

I strongly recommend publication in Nature communication.

Congratulation for this work.

Reviewer #3

(Remarks to the Author)

The authors did a good job in addressing my questions and comments.

I have no further issues with the paper.

Reviewer #4

(Remarks to the Author)

Reviewer #1 (Remarks to the Author):

This is an interesting article in a fast-moving area within nanophotonics. It merits publication since it provides topical new results that show how appropriate nanostructure can be used to improve the plasmonic performance of conducting polymers. In general the work is very well presented.

We are grateful for your appreciation of our study.

There are a number of questions that the authors should consider:

Main text

1) The lattice resonances that are considered here usually require a homogeneous (refractive index) environment if they are to be observed. Maybe I missed this, but I could not see in the manuscript whether a homogeneous environment had been used. If YES, then this needs to be stated, including the technique used; If NO, then do the authors have an explanation for why they see the resonances that they observe, despite the lack of a homogeneous environment?

We thank the reviewer for this comment which helped us to clarify the environmental conditions. In this work, we did not use a homogeneous (refractive index) environment. The extinction spectra of the PEDOT nanoantenna arrays were all measured in air on glass or CaF₂ substrate, and we did not use any additional dielectric materials or solvents to create a homogeneous environment. The reviewer is correct that collective lattice resonances (CLRs) often require a homogenous environment, which can circumvent negative effects from interface reflection and support radiative coupling between nanoantenna units. However, this is only a “soft” requirement to excite CLRs (Chem. Rev. 2018, 118, 5912–5951). For example, **systems with comparably large nanoantennas (and thereby large isolated polarizability) can diminish negative effects of reflection under inhomogeneous environments and enable CLRs to be observed under air-substrate conditions**, which is confirmed by previous reports (e.g. Appl. Phys. Lett. 2013, 102, 221116; Adv. Optical Mater. 2020, 8, 1901109).

To better understand the effect of environment for our PEDOT-based systems, we analyzed extinction spectra and $S-(1/\alpha_{iso})$ relationship for inhomogeneous and homogeneous environments (Figure R1). Figure R1a shows the results for a PEDOT nanoantenna array optimized for achieving CLR in a homogeneous environment (refractive index $n = 1.5$), exhibiting a sharp resonance ($\lambda_r = 2.27 \mu\text{m}$) with narrow linewidth $FWHM = 0.15 \mu\text{m}$. Changing from homogeneous to inhomogeneous environment for this nanoantenna array (Figure R1b) transforms the CLR into a broad localized surface plasmon resonance peak ($\lambda_r = 1.80 \mu\text{m}$, $FWHM = 0.8 \mu\text{m}$). However, this does not mean that a homogeneous environment is required for CLRs, because changing the environment also affects the matching conditions by red-shifting the isolated polarizability curves ($\text{Re}[1/\alpha_{iso}]$ and $\text{Im}[1/\alpha_{iso}]$). This can be seen from the bottom part of Figure R1b which shows that there is no longer CLR matching conditions in the inhomogeneous environment (no crossing points showing $\text{Re}[1/\alpha_{iso}] = S_r$ around $\lambda_r \approx n_s r$). Importantly, we can adjust the isolated polarizability to again reach matching conditions also for the inhomogeneous environment, by increasing the nanoantenna diameter (from 0.30 to 0.52 μm , Figure R1d). This **generates a narrow CLR at $\lambda_r = 2.27 \mu\text{m}$ ($FWHM = 0.3 \mu\text{m}$)**,

where CLR matching conditions are fulfilled. With the same geometric parameters of this nanoantenna array, the CLR conditions can be partially fulfilled for the homogeneous environment but with less optimized conditions (larger normalized detuning wavelength (Δ) and $|\text{Im}[1/\alpha_{\text{iso}}] - S_i|$), leading to less high-Q resonances for the homogeneous environment (Figure R1c) than for the inhomogeneous environment (Figure R1d).

The results confirm that **large-sized PEDOT nanoantennas can achieve CLR in the inhomogeneous environment**, and highlight that the nanoantennas (or period) needs to be re-optimized if changing the environment.

Figure R1. Analysis of array factors and simulated extinction spectra for square arrays (periodicity $r = 1.5 \mu\text{m}$) with inhomogeneous or homogeneous environments, based on

acid-treated PEDOT:ToS. The extinction spectra were obtained by FDTD simulations. (a) (b) Homogeneous environment (refractive index of the substrate and background both set to 1.5) and inhomogeneous environment (air-substrate), respectively, where the nanodisk diameter was set to 0.30 μm . (c) (d) Homogeneous and inhomogeneous environments, respectively, where the nanodisk diameter was increased to 0.52 μm . The height of all nanodisks were set to 0.2 μm .

Although immersed in air, we note that the PEDOT-based CLR actions still occur at the interface of the dielectric substrate region ($n = 1.5$). This is confirmed by angle-dependent spectra for which the effective refractive index can be fitted to around 1.5 (very approximate to the refractive index of glass substrate) in the angle-dependent equation of dipole radiation and diffraction grating effect (shown in Figure 4 in the manuscript).

In the revised manuscript, we have added the sentence “Considering the inhomogeneous environment consisting of dielectric substrate ($n_s = 1.5$) and air, we applied $\epsilon_s = 1.3$ in the calculation of α_{iso} ” in the first paragraph of the section called “Determining CLR matching conditions for PEDOT nanoantenna arrays”. We have also added the following sentence in the Methods section: “During all above extinction characterizations, the sample of PEDOT nanoantenna array was placed in air”. These sentences clarify to readers that this work focuses on conducting polymer CLR under inhomogeneous environment.

2) It would be an improvement if the authors mentioned the spectral region of interest sooner, ideally in the abstract, and earlier in the introduction.

Thank you for your suggestion. The revised manuscript now states the wavelength range for the resonances already in the abstract.

3) There are numbers missing from the axes in fig 1a (whilst the figure is illustrative, it would help the reader if they specified the spectral range)

As suggested, we have added the wavelength range for the spectra in Fig. 1a (shown in Figure R2 below). The reason to use arbitrary units for the y axis is that the two curves have different units, since the CLR is in units of percentage while the LSPR system corresponds to the optical cross-sectional area of a single nanoantenna.

Figure R2 (part of new Figure 1a). Extinction spectra of PEDOT nanoantennas without coupling effects (dashed purple line, simulated by the localized model without periodic boundaries) and with coupling through CLR (red full line).

4) When describing the enhancement in Q, it might be better to simply state the best Q obtained (12 I think) instead of "enhanced up to 10 times", "by around one order of magnitude" (conclusion).

Thank you for your suggestion. We have revised the text to focus on the obtained values, both in the abstract ("...with Q up to 12") and in the conclusion part ("from typical values below 1 to values above 10" and "with experimental Q-factors up to 12").

Supp Info

1) Lorentz oscillator parameters are given in many places. Please also provide the relevant Lorentz oscillator equations. If this is done then others can replicate the work.

Thank you for the suggestion. The Drude-Lorentz equation (with Lorentz oscillators) for fitting the permittivities (referred to the literatures: Nat. Nanotechnol. 2020, 15, 35-40; J. Mater. Chem. C, 2019, 7, 4350-4362) is described as:

$$\varepsilon(\omega) = \varepsilon_{\infty} - \frac{\omega_p^2}{\omega^2 + i\omega\gamma} - \sum_j \frac{A_j}{\omega^2 - \omega_j^2 + i\omega\gamma_j} \quad (\text{R1})$$

where ω is the angular frequency, ε_{∞} is the permittivity at infinitely high frequency (beyond the measurement range), γ is the momentum-averaged broadening (related to the damping relaxation), i is the imaginary unit, ω_p is the plasma frequency. A_j , ω_j and γ_j are amplitude, resonance angular frequency, and broadening (or damping relaxation part) of the j -th Lorentz oscillator, respectively. The Drude part has a similar mathematic type of Lorentz oscillator, although with $\omega_j = 0$.

We have added the above description in the Supplementary Information (Page 3).

2) A suggestion. On p.3, It seems odd that there is only one Drude component in-plane, whilst out-of-plane there are two Lorentz oscillators, at 0.025 eV and 0.028 eV, and these two oscillators seem to do the same job. Could these two out-of-plane LOs be replaced with a (single) Drude component?

We are grateful for this question. For the wavelength range of interest for this work, these two out-of-plane low energy LOs could indeed be replaced by a Drude oscillator ($\gamma = 0.329$ eV, $A = 0.128$ eV²), and such replacement also made the fitting good in the mid-infrared wavelength region. We have revised the parameters of the Drude-Lorentz models and refitted the spectroscopic ellipsometry based on this change, as shown in the Supplementary information (Supplementary Fig. 3 and Table 2, and updated calculations and simulations).

3) It looks as though the authors are trying to fit over the spectral region 0.14-5.8 eV, and yet have multiple oscillators with energy < 0.14 eV and multiple oscillators with energy > 5.8 eV. It's not unusual to have an oscillator beyond the region to which a fit is to be produced, however, could they achieve a similar fit with just one oscillator at each end of the spectrum?

We have followed this good suggestion and changed to use only one oscillator at each end outside the measured spectral range (marked by bold and underline in Table R1-R4). The permittivity at infinite frequency was also modified correspondingly to account for small deviations during the fitting. Inspired by the above suggestion and the previous comment, we also used a Drude oscillator to replace several Lorentz oscillators in the mid-infrared end of the spectrum (both for oxidized and reduced states of PEDOT). After the above adjustments, the fitting of the spectroscopic ellipsometry data could be slightly improved (updated in Supplementary Fig. 1, 2, 43 and 44). As the permittivities were slightly changed, we have rerun all FDTD simulations (covering Fig. 3a-3c, 4b, 4c, 4e, 4g and 4i in the manuscript and Supplementary Fig. 5, 7, 9, 11, 13, 15, 20, 36a, 37a, 38a, 39, 40 and 50) and dipolar coupling calculations (covering Fig. 2a-2c, 5e and 5f in the manuscript and Supplementary Fig. 5-16 and 20-21) using the new permittivity data (only minor changes to the results and no changes to the conclusions).

Table R1: Oscillators for oxidized state along the in-plane direction

$\epsilon_\infty = 2.310$			
Oscillator No. (j th)	Frequency ω_j (eV)	Broadening γ_j (eV)	Amplitude A_j (eV ²)
Drude	0	0.320	5.280
1	3.049	0.125	0.661
2	1.680	0.9827	1.242

Table R2: Oscillators for oxidized state along the out-of-plane direction

$\epsilon_{\infty} = 1.798$			
Oscillator No. (j th)	Frequency ω_j (eV)	Broadening γ_j (eV)	Amplitude A_j (eV ²)
Drude	0	0.329	0.128
1	8.000	3.932	6.291
2	4.532	0.604	0.125
3	0.711	1.050	0.248

Table R3: Oscillators for reduced state in the in-plane direction

$\epsilon_{\infty} = 1.100$			
Oscillator No. (j th)	Frequency ω_j (eV)	Broadening γ_j (eV)	Amplitude A_j (eV ²)
Drude	0	0.449	0.149
1	2.018	0.766	4.700
2	6.684	1.494	48.984

Table R4: Oscillators for reduced state in the out-of-plane direction

$\epsilon_{\infty} = 1.201$			
Oscillator No. (j th)	Frequency ω_j (eV)	Broadening γ_j (eV)	Amplitude A_j (eV ²)
Drude	0	0.329	0.0428
1	4.199	0.213	17.65
2	2.195	0.227	0.353
3	1.036	0.0749	0.0354

4) The fit in the region (up to 5 μ m) in supp figure S2 is not as good as the excellent fit in the NIR in supp figure S1; in particular the Psi spectra seem to be off by a couple of degrees, which in turn means that the optical constants will be slightly out. I don't know if it's enough to worry about, but won't a small change in optical constants have a bit of a knock-on effect in their coupled dipole calculations?

Thank you for the question. To test whether or not small changes in optical constants could have a knock-on effect, we have calculated $\text{Re}(1/\alpha_{\text{iso}})$ and $\text{Im}(1/\alpha_{\text{iso}})$ using a new permittivity dispersion denoted as ϵ_1 and compared the results with calculations based on the permittivity used in the Supplementary Information (here denoted ϵ_2 , based on the Drude-Lorentz model, shown in Table R1-R4). We made the new permittivity ϵ_1 more accurately fit the ellipsometry data in the wavelength range of 1.6-5.0 μ m (Figure R3) by fitting the in-plane direction through the Drude model (with broadening $\gamma = 0.32$ eV and amplitude $A = 5.28$ eV², **same parameters as shown in Table R1**) while fitting the out-of-plane direction by the B-Spline model (shown in Figure R3). This is different from ϵ_2 for which we strictly used the Drude-Lorentz model along both the in-plane and out-of-plane directions.

Figure R4 presents the resulting in-plane permittivity dispersions, showing very similar results for ϵ_1 and ϵ_2 . The results show that even with more accurate fitting, ϵ_1 and ϵ_2 (both along the in-plane direction) can be still very similar to each other. Along the in-plane direction, **there is no significant difference between ϵ_1 (more accurate fitting) and ϵ_2**

(slightly less good fitting in the Supplementary Information) if only slightly increasing the deviation. Next, we did calculations to ensure that the small difference between ϵ_1 and ϵ_2 would not induce noticeable differences in the isolated polarizability (Figure R5). Figure R5a-R5e shows that $\text{Re}(1/\alpha_{\text{iso}})$ from ϵ_1 closely overlaps with $\text{Re}(1/\alpha_{\text{iso}})$ from ϵ_2 , and $\text{Im}(1/\alpha_{\text{iso}})$ from ϵ_1 also largely overlapped with $\text{Im}(1/\alpha_{\text{iso}})$ from ϵ_2 . Furthermore, at the individual resonance wavelength (almost the same using ϵ_1 or ϵ_2), both the real and imaginary components of $1/\alpha_{\text{iso}}$ calculated using ϵ_1 are very similar to those calculated using ϵ_2 (with larger deviations of $1/\alpha_{\text{iso}} \leq 5 \mu\text{m}^{-3}$), which is shown in Figure R5f. Hence, we conclude that there is no knock-in effect and either set of permittivity dispersions would be ok to use without affecting the conclusions in the paper. In the updated manuscript, we decided to use ϵ_2 (fitting by strictly using Drude-Lorentz models) rather than ϵ_1 . Although ϵ_1 is obtained based on more accurate fitting (thanks to the B-spline model along the out-of-plane direction), **such almost perfect fitting can cause an overfitting problem in the sense that the noise is also fitted and generates artificial features in the permittivity** (as discussed in, e.g.: Thin Solid Films, 2017, 636, 519–526; Adv. Opt. Techn. 2022, 11(3–4), 93–115; Modeling Aspects in Optical Metrology VI, 2017, 103300B). In contrast, ϵ_2 that is achieved by strictly fitting with the Drude-Lorentz model **can balance between ensuring sufficiently good fitting and avoiding overfitting problems, and the slight deviation cannot affect the findings and conclusions.** Thus, we used ϵ_2 in this work.

Figure R3. Spectroscopic ellipsometry data (ranging from 1690 nm to 9000 nm) for the fitting of permittivity ϵ_1 of the PEDOT film (acid-treated PEDOT:Tos, with thickness of 200 nm). These raw data and the fitting processing are based on the VASE software. The Psi (ψ , marked in the red line) and Delta (Δ , marked in the green line) are achieved with three angles (50°, 60°, and 70°). The black dashed lines are shown in the best fitting data (using only Drude model along the in-plane direction, and using B-Spline model along the out-of-plane direction).

Figure R4. Permittivity for ϵ_1 and ϵ_2 along the in-plane direction. ϵ_1 part is fitted as shown in Figure R3, and ϵ_2 (with Lorentz oscillators) is the permittivity fitted previously in the Supplementary Information (with a slight deviation).

Figure R5. Analysis of array factors and extinction spectra for square arrays with different periodic distances $r = 1.2-1.6 \mu\text{m}$, by using ϵ_1 part (fitted shown in Figure R3) for acid-treated PEDOT:ToS. The S_r and $|\text{Im}(1/\alpha_{\text{iso}}) - S_t|$ both at CLR-matching conditions are shown in (f). ϵ_2 (with Lorentz oscillators) is the permittivity fitted previously in the Supplementary Information.

Reviewer #2 (Remarks to the Author):

This is a very interesting paper which clearly demonstrate that resonance strength and quality factor of conducting polymer plasmonic can be strongly improved through collective lattice resonance. The results are, in my opinion, very important and original. They solve a major bottleneck of organic plasmonics and widens the practical applications of organic plasmonics as building blocks of active metasurfaces and nano-optic systems.

We thank the reviewer for the appreciation of our study.

The manuscript is well written but I suggest to define more carefully the key parameters such as S (page 3) and or Δ (page 5) as many readers will not be specialized enough in lattice effect and detuning wavelength to easily follow the simulation page 4 and 5.

Thank you for the good suggestion. We have added the definitions and more explanatory meanings of S and Δ in the manuscript. For S , we have replaced simple reference to “lattice effects” to more detailed phrases that clarifies the role of dipole coupling (page 4):

“In periodic arrays, the polarizability of a single nanoantenna is transformed from the isolated α_{iso} to a periodic mode α_p through lattice-induced coupled dipole interactions (S), according to^{6, 34, 37}:

$$\alpha_p = (1/\alpha_{iso} - S)^{-1} \quad (1)$$

and S quantifies the sum of coupled dipole interactions (including electrostatic and radiative couplings^{38, 39}) generated from other nanoantenna units in the lattice and can be evaluated using^{34, 37}:

$$S = \sum_j^N \exp(ikr_j) \left[\frac{(1-ikr_j)(3\cos^2 \theta_j - 1)}{r_j^3} + \frac{k^2 \sin^2 \theta_j}{r_j} \right] \quad (2)$$

where r_j is the centre to centre distance between two nanoantennas, θ_j is the angle between the electrical field direction and the lattice vector direction, and N is the number of nanoantennas along the specific direction.

For Δ , it is used to evaluate the **wavelength difference between the surface plasmon resonance (λ_r) and the lattice-induced diffraction orders ($\lambda = n_s r$)** via the equation $\Delta = (\lambda_r - n_s r)/n_s r$. The smaller wavelength difference ($\Delta \rightarrow 0$) reflects the more significant contribution of nonlocal diffraction orders to the surface plasmon resonance, which serves as prerequisite for CLRs. We added this sentence in the manuscript (Page 5).

We hope that these revisions will help readers to better understand and appreciate these parameters.

Experimental results are impressive and will be of interest to many scientists reading nature communications.

I strongly recommend publication in nature communication

We are very grateful for your appreciation and recommendation, and hope that our manuscript will be interesting and useful for the readers of the journal.

Reviewer #3 (Remarks to the Author):

The manuscript titled "Switchable Narrow Nonlocal Conducting Polymer Plasmonics" describes how using a regular array (in either a hexagonal or square lattice) can increase the otherwise poor Q-factors of conducting polymer plasmonics. They also demonstrate how switching PEDOT between redox states can be used to switch on and off the plasmonic properties of the arrays.

The paper is well structured, and nicely combines extensive simulations and experiments. They show how the diameter and lattice spacing can be tuned to optimise the Q-factor of the structures, allowing q-factors to be reached around 10-12 at $\sim 4 \mu\text{m}$.

Thank you very much for your positive evaluation and kind appreciation of our study.

1. One of the notable results is the increase in q-factor as a result of the array though both the number of nearest neighbours and the ordered lattice is discussed. Since the hexagonal lattice and square lattice provide roughly similar results and the hexagonal lattice does a little better since it has "more adjacent neighbours" It would be interesting to disentangle local effects from periodic lattice effects. Can the author think of a way of disentangling adjacent neighbour effects from periodic lattice effects, e.g. what would happen in a similarly spaced but disordered array? How large does the lattice need to be to get this benefit?

Thank you for the questions. In our response, we first form a system with a localized surface plasmon resonance (LSPR) that is disentangled from lattice effects, for which **we designed and fabricated a nanoantenna array with a larger periodic distance of $r = 3.0 \mu\text{m}$** . This is 2-3 times larger periodicity than for the arrays with CLR's discussed in the manuscript, while the nanoantenna's dimensions are the same (the height of $0.2 \mu\text{m}$ and the diameter of $\sim 0.5 \mu\text{m}$, as shown in Figure R6a and R6b). Figure R6c shows that the array with $r = 3.0 \mu\text{m}$ generates only a weak and broad resonance peak at around $2.1 \mu\text{m}$, and the linewidth is $FWHM \approx 1.1 \mu\text{m}$, which is around 2.5 times broader than the CLR actions from the array with $r = 1.5 \mu\text{m}$. Moreover, the extinction for $r = 3.0 \mu\text{m}$ is only around half of that from $r = 1.5 \mu\text{m}$ even after compensating for the lower surface coverage for the larger periodicity. Figure R6d shows that such linewidth is similar to the localized surface plasmon resonances of such single antennas (from FDTD simulations without using periodic boundaries). Furthermore, **angle-dependent extinction spectra in Figure R6e and R6f (from both simulations and experiments) shows that the weak and broad surface plasmon resonance do not have the type of angle-dependent features that would be required for either diffraction grating effect or lattice dipole radiation coupling**. Therefore, such localized surface plasmon resonance can be disentangled from the lattice effect if under larger periodic distances $r = 3.0 \mu\text{m}$

Figure R6. LSPR from PEDOT-based periodic arrays. (a) (b) SEM images of nanoantenna arrays (diameter of nanoantenna units: $d \approx 0.51 \mu\text{m}$; height of nanoantenna units: $h = 0.2 \mu\text{m}$. periodicity: $r = 3.0 \mu\text{m}$). (c) Experimental extinction spectra of PEDOT-based periodic arrays with $r = 3.0 \mu\text{m}$ (marked in orange line, all on glass substrate) and $r = 1.5 \mu\text{m}$. Considering the number of nanoantenna units for $r = 1.5 \mu\text{m}$ is 2.25 times more than those for $r = 3.0 \mu\text{m}$, we multiply 2.25 with extinction intensity for $r = 3.0 \mu\text{m}$. (d) Comparison with the FDTD simulations of the localized models (without periodic boundaries, marked in blue line). (e) (f) Angle-dependent extinction spectra (Normalized) from simulations and experiments, based on p-polarization.

Figure R7. FDTD simulations on the investigation of adjacent neighbour effects within the $N \times N$ nanoantenna matrix. (a) Simulation models, where N is the nanoantenna number along the matrix side. Within the $N \times N$ nanoantenna matrix region, and the lattice corresponds to the case that N is infinite. **The center-to-center distance between two**

adjacent nanoantennas is about $r = 1.5 \mu\text{m}$, while such distance is set as **$2r = 3.0 \mu\text{m}$ between two nanoantenna units individually belonging to two different $N \times N$ nanoantenna matrices**. The absolute and normalized extinction spectra are shown in (b) and (c), respectively.

Next, we address the aspect of disentangling the effect of adjacent nanoantenna units from the lattice, and also the role of the size of the lattice. As shown in Figure R7a, we designed $N \times N$ nanoantenna matrices with different values of N . For each matrix, the center-to-center distance between two adjacent nanoantennas within each matrix was set to $r = 1.5 \mu\text{m}$ to ensure efficient radiative coupling interaction between neighboring units. The effective distance between neighboring nanoantennas individually belonging to two different matrix regions was set to **$2r = 3.0 \mu\text{m}$ to sufficiently avoid coupling** between such adjacent nanoantenna units individually belonging to two different nanoantenna matrix regions. In Figure R7b and R7c, **when enlarging the nanoantenna matrix from 1×1 to 2×2 , the incorporation of neighboring nanoantennas enhances the extinction intensity, and the surface plasmon resonance becomes slightly narrower, along with a slight red-shift of the resonance peak toward $\lambda_r = n_s r = 2.25 \mu\text{m}$. Further enlarging the matrix from 2×2 to 7×7 gradually makes the extinction peak more intense and narrower (*FWHM* up to $0.5\text{-}0.6 \mu\text{m}$)**, although it is still broader than the CLR if N is infinite. These results suggest that, **if without long-range lattice structures, only neighboring effects for closely adjacent nanoantenna units in the small matrix ($N < 5$) can make a clear but rather limited contribution to the narrowing of the resonance compared with the CLR with long-range order at large scale**. Likewise, some but limited CLR action could likely be achieved if introducing some degree of disorder into the lattice, since previous works showed that moderate degree of disorder (<30-40%) can still provide some lattice contribution (J. Chem. Phys. 2004, **121**, 12606).

Figure R8. Analysis of array factors (S) dependent on the number of nanoantenna units in the matrix. (a) S -($1/\alpha_{iso}$) relationship within the $N \times N$ nanoantenna matrix. S_r and S_i are the real and imaginary part of S , respectively. $\text{Re}(1/\alpha_{iso})$ and $\text{Im}(1/\alpha_{iso})$ are the real and imaginary part of $1/\alpha_{iso}$, respectively. (b) (c) S_r and S_i curves when further enlarging the $N \times N$ nanoantenna matrix, respectively. The center-to-center distance of nanoantenna units is set as $r = 1.5 \mu\text{m}$.

To gain further insight into the mechanism of changes in extinction spectra with increasing N of the matrix, we analytically calculated how the array factor (S) varies with the number of nanoantenna units in the matrix. Here, we use the equation:

$$S(N-1) = \sum_j^{N-1} \exp(ikr_j) \left[\frac{(1-ikr_j)(3\cos^2 \theta_j - 1)}{r_j^3} + \frac{k^2 \sin^2 \theta_j}{r_j} \right] \quad (3)$$

where N corresponds to the $N \times N$ nanoantenna matrix, indicating the dipolar coupling interactions generated from other nanoantenna units in the matrix. r_j is related to $(N-1)$ times periodic distances (r). Figure R8a shows that the maximum of S_r at the diffraction order wavelength ($\lambda = n_s r = 1.5 \times 1.5 \mu\text{m} = 2.25 \mu\text{m}$) gradually increases and sharpens with increasing N . For $N > 5$, S_r is likely sufficient to fulfill the requirement of $\text{Re}[1/\alpha_{iso}] = S_r$ in CLR-matching conditions. Meanwhile, S_i becomes more and more negative at $\lambda = n_s r$ when increasing N , which facilitates the suppression of damping relaxation confirmed by smaller values of $|\text{Im}[1/\alpha_{iso}] - S_i|$. **These results are consistent with the phenomena that the surface plasmon resonance becomes more intense and narrower when increasing numbers of nanoantenna units in the ordered matrix.** In Figure R8b and R8c, the S_r and S_i curves can be overlapped well if $N = 100$ - 1000 , which suggests that **essentially full benefit of CLRs should be achieved if there are at least 100-200 nanoantenna units along the matrix site ($N \geq 100$), corresponding to the size of at**

least 150-300 μm array. If N ranges from 5 to 100 (the sizes of 7.5-150 μm), we can still partly achieve the benefit of narrow resonances from CLRs, but such benefit is limited and compromised because the less negative S_i at $\lambda = n_s r$ is less efficient in suppressing damping relaxation through lattice effect.

We have fabricated PEDOT nanoantenna arrays to roughly confirm the above findings. For example, we prepared the 2×2 matrix described above, where the end-to-end distance of PEDOT nanoantenna in the matrix is $r = 1.5 \mu\text{m}$, but the effective distance between neighboring nanoantennas individually belonging to two different matrix regions was designed to $2r = 3.0 \mu\text{m}$ (as shown in Figure R9a and R9b). The large distance of 3.0 μm is sufficient for suppressing dipolar coupling interactions to adjacent nanoantennas, which is confirmed by the transformation into LSPR (verified in Figure R6). In Figure R9c, such small matrix exhibits a weak and broad resonance peak at 2.15 μm (the linewidth $FWHM \approx 1.05 \mu\text{m}$ and the quality factor $Q \approx 2$). What's more, Figure R9d do not show an angle-dependent feature for nonlocal resonance mode. Thus, **only with the closest neighboring effect from adjacent nanoantennas, the surface plasmon resonance from the 2×2 matrix system still primarily belongs to a localized mode.** To enhance such neighboring effect, we enlarged the matrix to the 4×4 pattern, where all geometric parameters are the same as the 2×2 pattern (shown in Figure R10a and R10b). This led to a slightly sharper resonance peak red-shifted to 2.25 μm ($FWHM \approx 0.9 \mu\text{m}$ and $Q \approx 2.5$, Figure R10c). Moreover, the angle-dependence of this resonance, assigned to the coupling with $(0, \pm 1)$ RA, is slightly improved (Figure R10d). These results suggest that **increasing the number of adjacent nanoantennas enhances dipole radiation and supports the transformation from localized to nonlocalized resonance modes**, which agree with the theoretical calculation that the more adjacent nanoantennas facilitate the more negative S_i values to suppress damping relaxation rates (Figure R8a). Even so, the weak extinction and the broad linewidth of surface plasmon resonance from the 4×4 matrix are still approximate to the localized mode.

To testify whether the lattice enables efficient CLR when N is in the ranges of 100-200 (corresponding to the size of 150-300 μm), we extended the matrix to 156×156 pattern (approximately at the middle of the N range of 100-200), which is characterized in Figure R11a and R11b. In Figure R11c, the extinction spectrum exhibits a narrower resonance peak at 2.26 μm . The angle-dependent extinction spectra in Figure R11d illustrate that such resonance is efficiently coupled with $(0, \pm 1)$ RA, confirming the presence of significant dipolar radiation. These observations verify the nonlocal resonance feature from the 156×156 matrix, which is almost the same as those in CLR features. Therefore, **the size ranging from 150 to 300 μm ($N = 100-200$) can be sufficient to achieve efficient benefit from CLRs.**

Figure R9. Structures of 2×2 PEDOT nanoantenna matrix and its extinction spectra. (a) (b) Images observed by optical microscopy and SEM, respectively. (c) Experimental extinction spectra under the normal radiation. (d) Angle-dependent extinction spectra (Normalized) from experiments, based on p-polarization. The end-to-end distance of PEDOT nanoantenna in the matrix is $r = 1.5 \mu\text{m}$, and the effective distance between neighboring nanoantennas belonging to two different matrix regions was set to $2r = 3.0 \mu\text{m}$. Each nanoantenna has a diameter of $0.5 \mu\text{m}$ and a height of $0.2 \mu\text{m}$.

Figure R10. Structures of 4×4 PEDOT nanoantenna matrix and its extinction spectra. (a) (b) Images observed by optical microscopy and SEM, respectively. (c) Experimental extinction spectra under the normal radiation. (d) Angle-dependent extinction spectra (Normalized) from experiments, based on p-polarization. The end-to-end distance of PEDOT nanoantenna in the matrix is $r = 1.5 \mu\text{m}$, and the effective distance between neighboring nanoantennas belonging to two different matrix regions was set to $2r = 3.0 \mu\text{m}$. Each nanoantenna has a diameter of $0.5 \mu\text{m}$ and a height of $0.2 \mu\text{m}$.

Figure R11. Structures of 156×156 PEDOT nanoantenna matrix and its extinction spectra. (a) (b) Images observed by optical microscopy and SEM, respectively. (c) Experimental extinction spectra under the normal radiation. (d) Angle-dependent extinction spectra (Normalized) from experiments, based on p-polarization. The end-to-end distance of PEDOT nanoantenna in the matrix is $r = 1.5 \mu\text{m}$, and the effective distance between neighboring nanoantennas belonging to two different matrix regions was set to $2r = 3.0 \mu\text{m}$. Each nanoantenna has a diameter of $0.5 \mu\text{m}$ and a height of $0.2 \mu\text{m}$.

Regarding changes to the revised manuscript and Supplementary Information, we have added results for periodic arrays with larger periodic distances $r \approx 3.0 \mu\text{m}$ to disentangle CLRs from localized surface plasmon resonance for the same sized nanoantennas. For example, we have added the sentence “If the periodicity is $r = 3.0 \mu\text{m}$, the linewidth of the surface plasmon resonance at $\lambda_r \approx 2.1 \mu\text{m}$ becomes much broader ($FWHM = 1.1 \mu\text{m}$ and $Q \approx 1.9$), which is almost the same as features of LSPR ($Q \approx 2$) observed by FDTD simulations”, in the first paragraph of “Experimental demonstration of narrow conducting polymer CLRs” (Corresponding SEM and extinction spectra have been added as Supplementary Fig. 27). For angle-dependence for this local mode, we have added the sentence “However, if the periodicity is $r = 3.0 \mu\text{m}$, the broad peak at $\lambda_r \approx 2.1 \mu\text{m}$ does not have the type of angle-dependent features that would be required for either diffraction

grating effect or lattice dipole radiation coupling, which confirms the LSPR feature that is disentangled from lattice effects”, in the second paragraph of “Understanding conducting polymer CLR through angle-dependence” (Corresponding angle-dependent extinction spectra are shown in Supplementary Fig. 40 in Supplementary Information).

Regarding array size effects, we have added information to the revised submission based on the calculation of array factors related to Figure R8 as new Supplementary Figure (corresponding to Supplementary Fig. 4) to predict the sizes that are small but enough to support efficient CLR like in the long-range periodicity, and such experimental confirmation ($N \approx 150$) is also added as Supplementary Fig. 42. Moreover, the experimental results about 2×2 matrix, that serve as the simplest model to confirm the adjacent neighbor effects, were added as Supplementary Fig. 41. In the manuscript, we briefly added the description of experimental results related to both models in the second paragraph of “Understanding conducting polymer CLR through angle-dependence” (Page 9). we revised the definition of N to “the number of other nanoantennas along the specific direction” (in Page 3), and we state “ $N = 1000$ ” for the calculation of S at various periodicity conditions (in Page 4).

2. They demonstrate the switching of the PEDOT arrays using either PEI or HCL vapor. In figure 5c they show the switching result for multiple REDOX cycles. Can the authors also include the off (reduced) state for each cycle in this figure?

Thank you for the comment. We have followed the suggestion and added the normalized extinction spectra for the reduced state for 1st ~ 4th redox cycles in Figure 5c (Figure R12a below). The absolute extinction spectra (Figure R12b below) are shown in Supplementary Information (Supplementary Fig. 47).

Figure R12. Measured extinction spectra (normalized) of the PEDOT nanoantenna array during the redox cycles. (a) the normalized extinction spectra that is corrected in the Fig. 5c in the manuscript. (b) the absolute extinction spectra that is added in Supplementary Fig. 47.

3. In the synthesis part it is not clear what is meant by: "the oxidant-coated films were exposed to EDOT vapors, generated by heating liquid EDOT at 60 °C, and then reacted

with EDOT for 40 min" Is that a EDOT solution, are any steps involved in between? or just dipped in pure EDOT? Is this based on a literature method?

Thank you for the question. In the step "reacted with EDOT for 40 min", the EDOT is "EDOT vapor", which can be transformed from pure EDOT liquid under vacuum. It is based on corresponding literatures: *Adv. Mater.* **33**, 2102451 (2021); *J. Mater. Chem. C* **7**, 4350-4362 (2019); *J. Mater. Chem. A*, 2018, 6, 21304–21312.

To make readers understand more, we have added the conditions "by heating liquid EDOT at 60 °C under vacuum" and "reacted with EDOT vapor" in the Methods ("Thin-Film Deposition").

Reviewer #4 (Remarks to the Author):

Thank you for co-reviewing.